# MultiOrg: A Multi-rater Organoid-detection Dataset

**Christina Bukas**[1], **Harshavardhan Subramanian**[1], **Fenja See**[2], **Carina Steinchen**[2]
**Ivan Ezhov**[3], **Gowtham Boosarpu**[4], **Sara Asgharpour**[2], **Gerald Burgstaller**[2]
**Mareike Lehmann**[2,4], **Florian Kofler**[1,5,6], **Marie Piraud**[1]

[1]Helmholtz AI, Computational Health Center (CHC), Helmholtz Munich, Neuherberg, Germany
[2]Institute of Lung Health and Immunity (LHI), Comprehensive Pneumology Center (CPC),
Helmholtz Munich, Member of the German Center for Lung Research (DZL), Neuherberg, Germany
[3]Technical University of Munich, School of Computation, Information and Technology,
Department of Computer Science, Munich, Germany
[4]Institute for Lung Research, Philipps-University Marburg, Universities of Giessen and
Marburg Lung Center, Member of the German Center for Lung Research (DZL), Marburg, Germany
[5] Department of Neuroradiology, Technical University of Munich, Munich, Germany.
[6] Department of Quantitative Biomedicine, University of Zurich, Switzerland.
`{christina.bukas,mareike.lehmann,marie.piraud}@helmholtz-munich.de`

## Abstract

High-throughput image analysis in the biomedical domain has gained significant attention in recent years, driving advancements in drug discovery, disease prediction, and personalized medicine. Organoids, specifically, are an active area of research, providing excellent models for human organs and their functions. Automating the quantification of organoids in microscopy images would provide an effective solution to overcome substantial manual quantification bottlenecks, particularly in high-throughput image analysis. However, there is a notable lack of open biomedical datasets, in contrast to other domains, such as autonomous driving, and, notably, only few of them have attempted to quantify annotation uncertainty. In this work, we present *MultiOrg* a comprehensive organoid dataset tailored for object detection tasks with uncertainty quantification. This dataset comprises over 400 high-resolution 2d microscopy images and curated annotations of more than 60,000 organoids. Most importantly, it includes three label sets for the test data, independently annotated by two experts at distinct time points. We additionally provide a benchmark for organoid detection, and make the best model available through an easily installable, interactive plugin for the popular image visualization tool Napari, to perform organoid quantification.

## 1 Introduction

Accurate and efficient object detection methods in biomedical image analysis are crucial for research and diagnostics. Designing such methods requires diverse, well-curated datasets of high-resolution images reflecting real-world complexities. The annotation of biomedical datasets represents a labor-intensive and subjective process relying on human experts. This work represents a multi-rater organoid dataset designed for benchmarking object detection algorithms in a label-noise-aware setting that embraces the subjectivity in labels.

Organoids are miniature three-dimensional (3d) models of organs grown in vitro from stem cells. They mimic the complexity and functionality of real organs, making them extremely valuable for medical research, disease modeling, and drug testing (Barkauskas et al., 2017; Kim et al., 2020; Ingber, 2022). Organoid cultures, deriving from healthy and diseased or genetically engineered cells and undergoing different conditions and treatments, can be grown for several months (Youk et al.,

2020; Huch and Koo, 2015). These high-throughput experiments are monitored via microscopic imaging and, therefore, necessitate fast and objective detection, quantification, and tracking methods (Rios and Clevers, 2018; Du et al., 2023). Detection of organoids in real-world lab-culture images is associated with many challenges (Kassis et al., 2019). Beyond the typical challenges associated with microscopy (out-of-focus, lightning, padding, etc...), those 3d cultures are imaged in 2d, leading to overlapping structures. Organoids can highly vary in size, shape, and appearance (Domènech-Moreno et al., 2023), and be difficult to distinguish from dust and debris present in the culture (Matthews et al., 2022; Keles et al., 2022). Finally, the high number of objects to analyze per image poses a big hurdle to a human prone to distraction and fatigue (Haja et al., 2023). Manual annotation of this data, which is still state-of-the-art (Costa et al., 2021; Wu et al., 2022) is, therefore, error- and bias-prone, which introduces noise in the labels. However, evaluating learning algorithms for organoid detection, involves comparing predicted outcomes to those manual annotations or '*Ground Truth (GT)*' during training and testing. As shown in our previous work, deep learning algorithms can outperform even highly-trained human annotators (Kofler et al., 2021). In complex real-life datasets, understanding the shortcomings that label uncertainty creates in the '*GT*' is, therefore, pivotal before training and benchmarking *Deep Learning (DL)* models. Moreover, quantifying the label noise by assessing the intra- and inter-rater reliability is crucial to interpret similarity metrics between model predictions and reference annotations (Kofler et al., 2023).

In this work, see Figure 1, we release *MultiOrg*, a large multi-rater 2d microscopy imaging dataset of lung organoids for benchmarking object detection methods. The dataset comprises more than 400 images of an entire microscopy plate well and more than 60,000 annotated organoids, deriving from different biological study setups, with two types of organoids growing under varying conditions. Most importantly, we introduce three unique label sets derived from the two annotators at different times, allowing for the quantification of label noise (see Fig. 2). Such a dataset can enable the community to explore biases in annotations, investigate the effect these have on model training, and promote the active area of research for uncertainty quantification. To our knowledge, this is the second largest organoid dataset to date to be made freely available to the community (Bremer et al., 2022). It is also the first organoid dataset and one of the very few biomedical object-detection datasets to introduce multiple labels (Nguyen et al., 2022; Amgad et al., 2022). We benchmarked this dataset by training and testing four widely established *DL* models for object detection tasks using both one-stage and two-stage architectures. Finally, along with the dataset and model, we release a tool for quantifying lung organoids, enabling users to visualize and correct the detected organoids before extracting useful features for downstream tasks. This tool solves the bottleneck of manual quantification of lung organoids, enabling high-throughput image analysis for biological studies.

In summary, the contributions of this work are as follows:

- We release *MultiOrg*, an object detection bio-medical dataset of more than 400 microscopy images comprising around 60,000 lung organoids annotated by two expert annotators.

- We provide quantification of label uncertainty through a Kaggle benchmark challenge that evaluates the submissions on the different test label sets.

- We benchmark our dataset on four standard object detection methods, show how performance varies depending on the selected annotations, and release the models on zenodo.

- We release the best model in a napari plugin, *napari-organoid-counter* (Bukas, 2022), which allows users to curate predictions, thus enabling high-throughput analysis.

## 2 Related work

Kassis et al. (2019) proposed *OrganoQuant*, a manually-annotated, human-intestinal-organoid dataset of around 14,000 organoids, along with an object detection pipeline based on Faster R-CNN Ren et al. (2015), to locate and quantify human intestinal organoids in brightfield images. Though object detection performance is satisfactory and the quantification process is robust, inference is performed on cropped patches of a well. Similarly, Matthews et al. (2022) released a dataset of brightfield and phase-contrast microscopy images and proposed an image analysis platform, *OrganoID*, based on U-Net Falk et al. (2019), which segments and tracks different types of organoids. They trained their model on images of pancreatic cancer organoids and validated it on pancreatic, lung, colon, and adenoid cystic carcinoma organoids. This work introduces several types of organoids. However, the

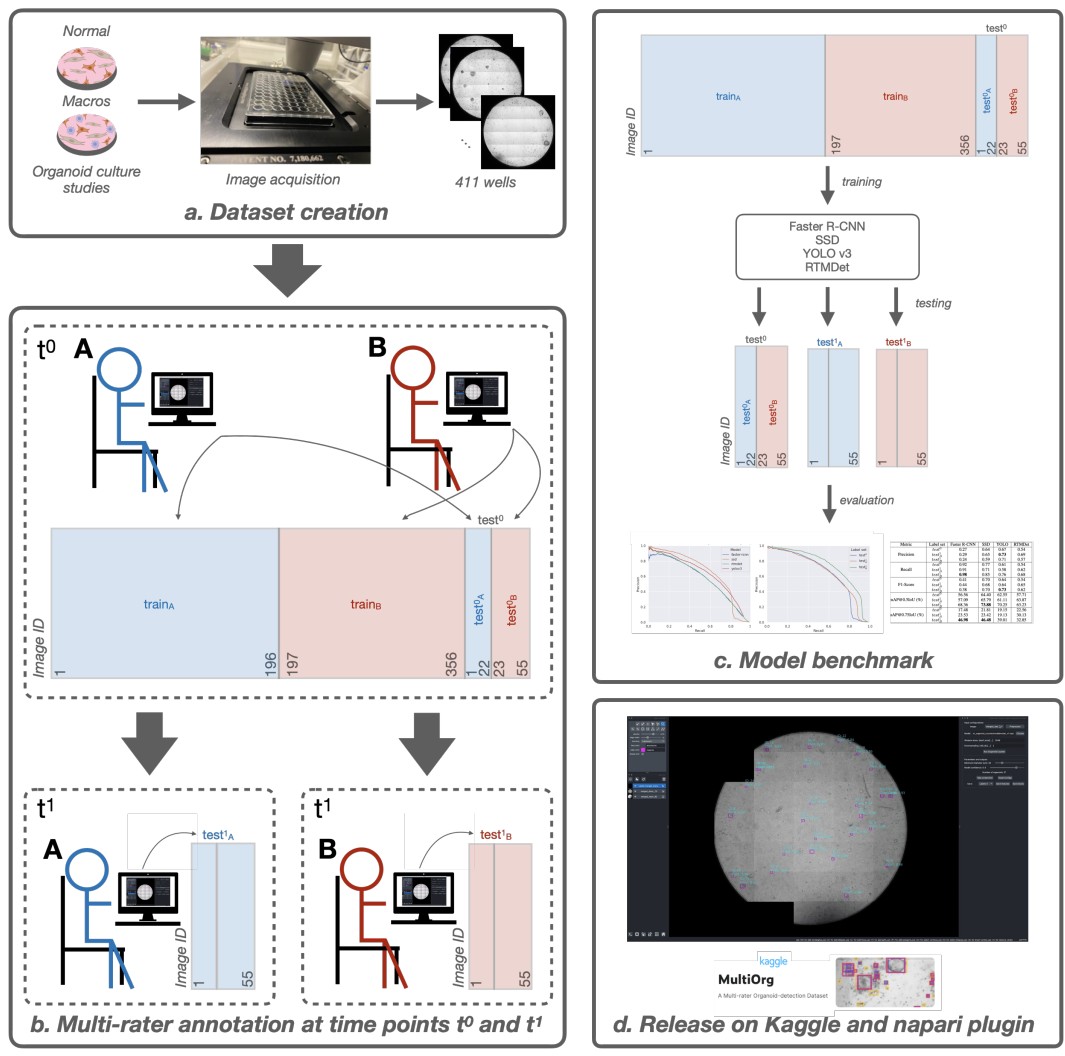

Figure 1: *MultiOrg* workflow. a) Dataset creation, b) Multi-rater annotation at time points $t^0$ and $t^1$, c) Model benchmark, and d) Release on Kaggle and napari plugin

dataset is small, including only 66 images featuring 5 to 50 organoids each. In Haja et al. (2023), *OrganelX* platform was released to enable segmentation of murine liver organoids using Mask-RCNN He et al. (2017). Furthermore, Bian et al. (2021) introduced a high-throughput image dataset of liver organoids for detection and tracking. They also propose a novel deep neural network architecture to track organoids dynamically and detect them quickly and accurately. However, here, too, the dataset size is relatively small, with 75 images containing a total of 6,482 organoids. Bremer et al. (2022) used a multicentric dataset consisting of 729 images containing 90,210 annotated organoids, including multiple organoid systems like liver, intestine, tumor, and lung, and proposes an organoid annotation tool, *GOAT*, which uses Mask R-CNN (He et al., 2017), for unbiased quantification. The corresponding dataset contains six organoid types, generated in four centers and acquired with five microscopes. More recently, Domènech-Moreno et al. (2023) proposed an object detection algorithm, based on YOLO v5 Ultralytics (2021), *Tellu*, to classify and detect intestinal organoids of different types. The tool also enables automated analysis of intestinal organoid morphology and fast and accurate classification of organoids.

*MultiOrg* is, therefore, the second largest organoid dataset (see Table 1). It is noisier than those introduced above; it is not the densest but contains clumps of organoids and displays an extensive range of sizes, presenting one of the most challenging settings for object detection. We introduce

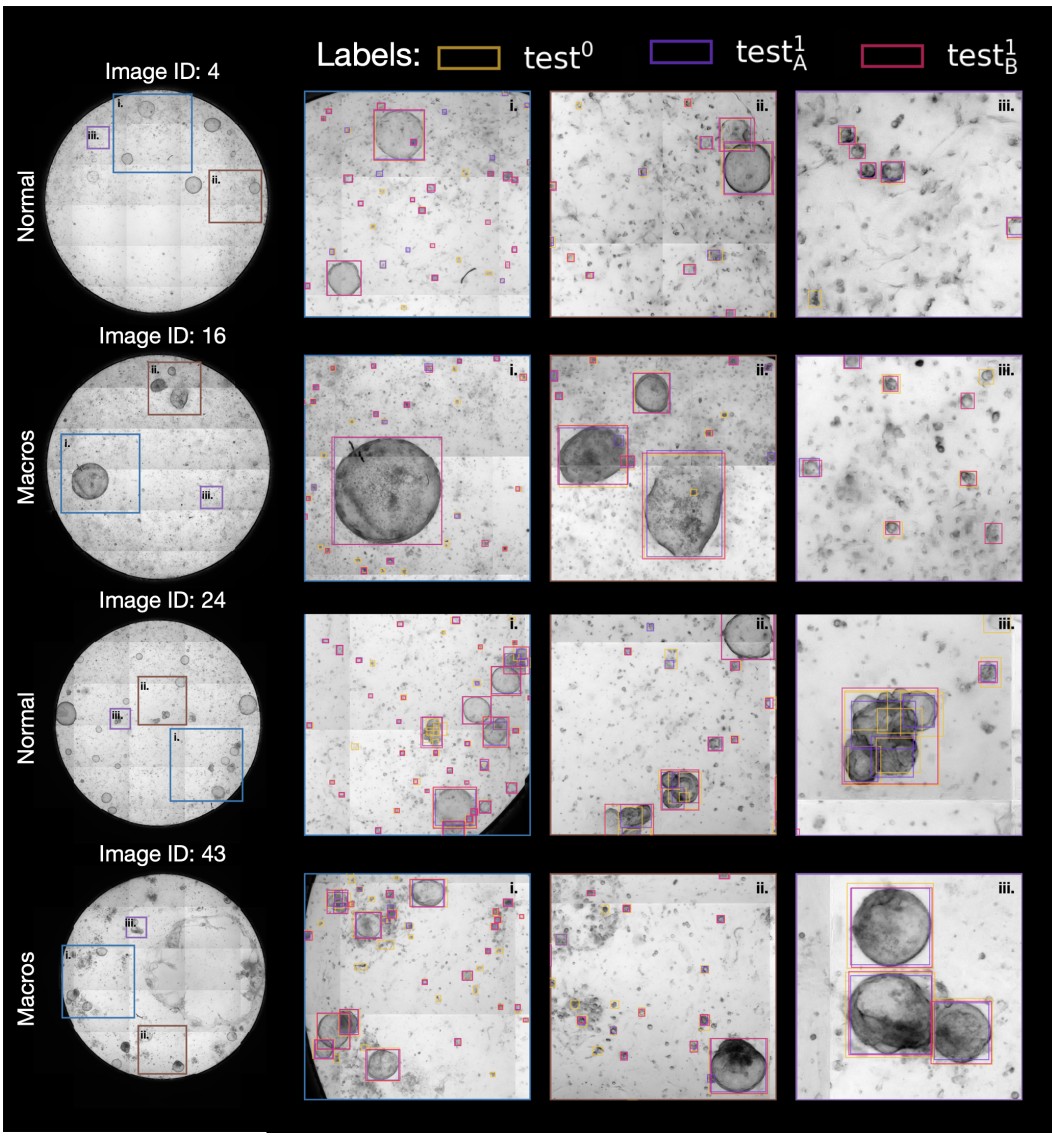

Figure 2: Multiple label sets in *MultiOrg*. Full test image (left) and crops of areas A, B, and C overlaid with $test^0$, $test^1_A$ and $test^1_B$ (right). The square crops are of sizes 1800, 1200, and 500 px. $test^0$ in images 4 and 16 (respectively 24 and 43) originates from Annotator A (resp. B). 'Macros' are typically noisier, as the cultures initially contain more cells (Appendix A.1.1).We observe a reduction in the number of annotations at time $t^1$, as the annotators do not consider some small organoids that were annotated at $t^0$. In image 24, Annotator B annotates clumps of organoids as one large object at $t^1$. The large structure in image 43 is an experimental matrigel artifact. The image-wise intra-rater Recall scores are 0.776, 0.532, 0.667 and 0.503 for images 4, 16, 24, and 43, respectively (with $test^0$ as *GT*).

Table 1: Overview of the published organoid datasets. We report the organs from which the organoids derive, the number of images, their resolution, the total number of annotated organoids, and the presence of multiple label sets in the dataset. *MultiOrg* is the only one to provide multiple label sets.

| Dataset | Organ | # Images | Image Resolution | # Organoids | Multi-Label |
|---|---|---|---|---|---|
| OrgaQuant (Kassis et al., 2019) | Intestine | 1750 | 300x300, 450x450 | 14,240 | × |
| OrganoID (Matthews et al., 2022) | Pancreas, Lung, Colon, Adenoid | 66 | 512x512 | 5-50/image | × |
| Bian et al. (2021) | Liver | 75 | 7227x7214 | 6,482 | × |
| GOAT (Bremer et al., 2022) | Liver, Intestine, Tumor, Lung | 729 | 512x512 | 90,210 | × |
| Tellu (Domènech-Moreno et al., 2023) | Intestine | 840 | 960x1280 | 23,066 | × |
| *MultiOrg* (ours) | Lung | 411 | 6390x5724 | 63,042 | ✓ |

several label sets on the test set to address this complexity. In the dataset, we focused on the detection task only, since it suffices for most practical applications and it is the challenging part from a machine learning point-of-view. Once the detection has been done, the segmentation can be obtained from pre-trained segmentation models (e.g., SAM Kirillov et al. (2023)).

None of the above-mentioned datasets related to the study of organoids in computer vision offer more than one set of labels. Nevertheless, comparing multiple annotations in *DL* is not new. Various previous initiatives have publicly released multi-rater biomedical datasets for image segmentation (Armato III et al., 2011; Styner et al., 2008; Almazroa et al., 2017; Lesjak et al., 2018; Mehta et al., 2022; Bran et al., 2024) and classification (Orlando et al., 2020; Sivaswamy et al., 2015; Aung et al., 2015). Fewer are available, though, for object detection. To our knowledge, two medical imaging datasets are currently available (Nguyen et al., 2022; Amgad et al., 2022). The *VinDr-CXR* dataset consists of 18k chest X-ray images annotated with bounding boxes by three radiologists for the presence of 28 lung diseases (Nguyen et al., 2022). The *NuCLS* dataset provides 97,000 annotations by 32 raters of nuclei from breast cancer pathology images (Amgad et al., 2022). Since labeling uncertainty in object detection is as common as in other image analysis tasks, we hope our dataset will help mitigate the gap in multi-rater detection datasets and contribute to advancing models embracing label variability.

## 3 Dataset

### 3.1 Dataset creation

*MultiOrg* consists of 411 bright-field microscopy images representing entire wells of lung organoids derived from murine cells and collected from 26 different studies. Each study can belong to one of two different types, either 'Normal' or 'Macros' (Figure 1(a) and Appendix A.1.1). During image acquisition, each 3d plate well was imaged in two-dimensional (2d) layers, each divided into smaller tiles (Appendix A.1.2). Individual tiles were then stitched together to form one stack of images. Since organoids are spherical structures, we applied maximum projection to merge this stack into a single plane, thereby reducing the annotation effort to one image per well, later estimating the organoid volumes from their 2d projection.

For dataset annotation, all organoids present in the images should be fitted by a bounding box. The annotation process was carried out as in Kastlmeier et al. (2023) by using the initial release, *v.0.1.0*, of the *napari-organoid-counter* tool (Bukas, 2022) to generate pseudo labels as a starting point for both annotators with a fixed set of parameters (see Appendix A.1.3). The dataset was initially annotated at time point $t^0$ by two annotators (see Figure 1(b)), namely Annotator A (53% of the images) and Annotator B (47% of the images). The images were then split into train and test sets stratified by annotators and study type. The training set then consists of 356 images derived from 25 studies, and the evaluation was performed on the remaining 55 images from 7 studies as a held-out test set (details in Table A.4). At time point $t^1$, Annotators A and B reannotated all test images, blinded to their initial labels (Figure 2). Annotator A used the same setup, whereas Annotator B changed their setup (computer mouse and monitor) between $t^0$ and $t^1$. The number of organoids annotated by each annotator, in each study type, for the train and test sets are provided in Table 2 and statistics on bounding box sizes in Figure A.5 and Table A.5.

We, therefore, provide three label sets for the images of our test set. The annotations produced at time point $t^0$ are denoted $test^0$ and can further be split into subsets $test^0_A$ and $test^0_B$ since images 1-22 were annotated by A and 23-55 by B (see Figure 1). Additionally, label sets $test^1_A$ (respectively

$test_B^1$) refer to the re annotation from annotators A (resp. B), at time point $t^1$ on all 55 images of the test set.

Table 2: Overview of the label sets (*train*, *test⁰*, $test_A^1$, and $test_B^1$). Number of images and organoid labels stratified by study type (for all) and annotator (only relevant for *train* and *test⁰*). All *test* label sets refer to the same images. We see a reduction in the number of labels between $t^0$ and $t^1$.

| Study Type | Normal | | Macros | | Combined | |
|---|---|---|---|---|---|---|
| | # Images | # Organoids | # Images | # Organoids | # Images | # Organoids |
| **Train set** | | | | | | |
| $train_A$ | 181 | 30,710 | 15 | 2,669 | 196 | 33,379 |
| $train_B$ | 135 | 20,263 | 25 | 1,781 | 160 | 22,044 |
| **Total** | **316** | **50,973** | **40** | **4,450** | **356** | **55,423** |
| **Test set** | | | | | | |
| $test_A^0$ | 8 | 1,145 | 14 | 1,865 | 22 | 3,010 |
| $test_B^0$ | 20 | 3,020 | 13 | 1,493 | 33 | 4,513 |
| **Total (Label set *test⁰*)** | **28** | **4,165** | **27** | **3,358** | **55** | **7,523** |
| **Label set $test_A^1$** | **28** | **2,748** | **27** | **1,981** | **55** | **4,729** |
| **Label set $test_B^1$** | **28** | **2,655** | **27** | **2,301** | **55** | **4,956** |

## 3.2 Object detection metrics

We compare the multiple label sets and assess the quality of model predictions using several evaluation metrics. *True Positives (TPs)*, *False Positives (FPs)*, and *False Negatives (FNs)* are computed for each image, for a given *Intersection-over-Union (IOU)* threshold, using one of the label sets as the 'GT'. Their total numbers are then aggregated on the entire test set. For comparing the three available label sets, we compute Precision and Recall at an *IOU* of 0.5 and the F1-score. While Precision measures the percentage of correct predictions against a considered true label, Recall (i.e., sensitivity) measures the proportion of true positive predictions identified correctly. For evaluating model performance, we use the *Precision-Recall (P-R)* curves as the primary tool. It consists of Precision and Recall values at different model confidence thresholds, at a fixed *IOU* threshold (here we use 0.5 unless specified otherwise). We also report *Mean Average Precision (mAP)*, by integrating precision across Recall levels from 0 to 1. We compute them using the standard library (Padilla et al., 2021) which follows the PASCAL VOC challenge technique for interpolating points on the curve (Everingham et al.).

## 3.3 Multi-rater analysis

We compute inter- and intra-rater uncertainties to quantify annotation variance and assess the consistency of the two raters over time and against each other. Inter-rater scores assess the inconsistency in the assessments made by different raters when evaluating the same image and can permit the detection of biases and different expertise levels. The variability in assessments made by one rater when evaluating the same image multiple times (intra-rater) can permit the detection of errors and ambiguities associated with the complexity of the task.

Figure 3 shows intra-rater scores, with label set $test^0$ used as the *GT* (note that switching the choice of *GT* does not impact the F1-score and switchs Recall and Precision). For Annotator A, we compare the annotations for test images 1-22, i.e., $test_A^0$ with the corresponding subset of $test_A^1$, while for Annotator B we compare annotations of images 23-55, i.e., $test_B^0$ with the corresponding subset of $test_B^1$. We find that Annotator A is more consistent over time, especially for the 'Normal' studies, which is in line with the reported change of annotation setup of Annotator B. Additionally, annotation of 'Macros' seems more challenging than 'Normal' images. We also find a reduction in the number of annotations at time point $t^1$, already reported in Table 2. The qualitative inspection of Figure 2 suggests that some pseudo labels used as a starting point for all manual annotations were not removed in $test^0$, probably indicating improvement of the annotations over time. Further comparison of the pseudo labels to the label sets can be seen in Table A.9. We also observe that Annotator B, unlike Annotator A, annotated overlapping clumps of organoids differently at $t^1$, which could be the source of the higher reported inconsistency.

Inter-rater scores can be computed for the entire test set between $test_A^1$ and $test_B^1$, as shown in Figure 3. We again observe that the annotation of 'Macros' is generally more challenging, and those images consistently appear noisier in Figure 2. In Figure A.6(bottom), we also display the inter-rater scores on the image subsets 1-22 and 23-55, as well as across $t^0$ and $t^1$. This corroborates the larger evolution of Annotator B between $t^0$ and $t^1$ and indicates a convergence of the two annotation styles. Table A.6, Table A.7, and Table A.8, provide all statistics for these multi-rater scores.

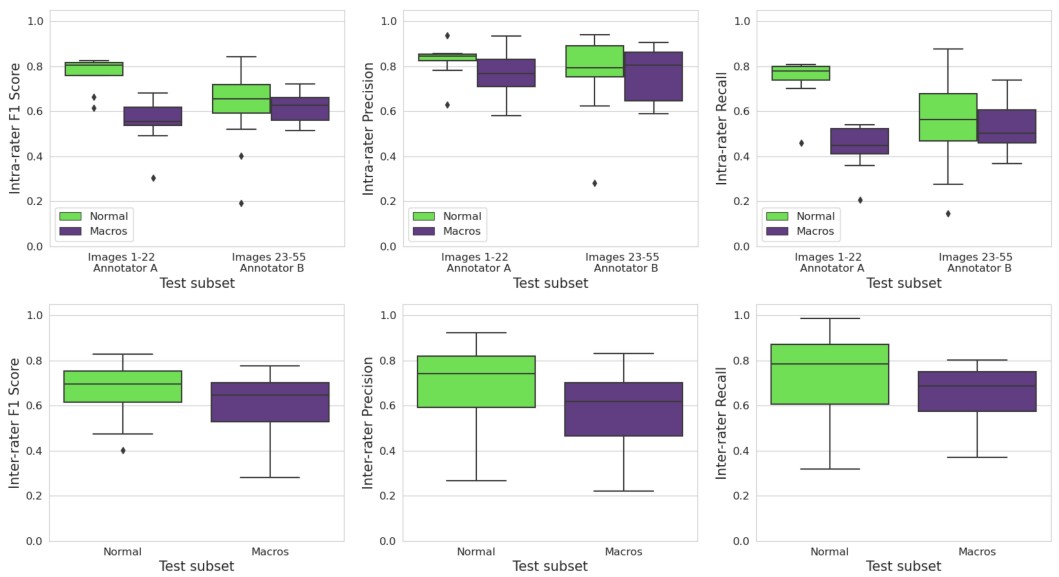

Figure 3: Multi-rater scores. **Top**: Intra-rater F1-score (left), Precision (middle), and Recall (right), where $test^0$ is considered the *GT*, for both annotators and according to study type. Annotator A appears more consistent on 'Normal' images (higher scores), and annotation of 'Macros' seems more challenging (with lower scores). Both annotators show an overall higher Precision and lower Recall, indicating that $test^0$ has many more annotations which are treated here as *FNs*. **Bottom**: Inter-rater F1-score (left), Precision (middle), and Recall (right) on the test set between $test_A^1$ and $test_B^1$, where $test_A^1$ is considered the *GT*, split according to study type. Raters agree more on 'Normal' images, indicating that the annotation of 'Macros' images is more challenging. Individual differences are generally lower than in-between raters (lower inter-rater than intra-rater scores).

### 3.4 Dataset availability

We make *MultiOrg* available to the community. All images are public on Kaggle, together with label sets $train$ and $test^0$, to ensure that the steps presented in Section 4 can be reproduced. The label sets $test_A^1$ and $test_B^1$ can be queried by participating in the *MultiOrg* challenge, where our leaderboard returns the average of *mAP* on $test_A^1$ and $test_B^1$. We invite scientists to participate, to promote research in the field of uncertainty estimation.

## 4   Model Benchmarking

We benchmark four standard object-detection *DL* models on *MultiOrg*:

- *Faster R-CNN* (Ren et al., 2015)

- *Single Shot MultiBox Detector (SSD)* (Liu et al., 2016)

- *You Only Look Once, Version 3 (YOLOv3)* (Redmon and Farhadi, 2018)

- *Real-Time Models for object Detection (RTMDet)* (Lyu et al., 2022).

All trained models can be found on zenodo, and code and documentation to reproduce the training is available on Kaggle [1].

## 4.1  Training and Testing

For training, the images in the training set were split into patches of 512x512 px, resulting in a total of 20,011 patches. Bounding boxes extending beyond the borders of the patches were omitted in the *GT* since these organoids can be captured by a sliding window approach at inference, resulting in 44,418 bounding box labels for training. For training and validation, we used the *mmdetection* (Chen et al., 2019) toolbox, with the original configuration for each model adapted such that the input and parameters for all models is the same (details in Appendix A.2).

During testing, sliding window inference was performed on the full images of the test set. We slide over each image twice with different window sizes and down-sampling factors to detect both small and large organoids. We empirically choose the following parameters: window size set to 512 and 2048 px, while the down-sampling factor is set to two and eight, respectively, with a window overlap of 0.5 and *Non max suppression (NMS)* for post-processing with a threshold of 0.5. For each model, we choose the checkpoint with the highest *mAP* on $test^0$, thus using this label set for validation during training. We report those, along with training and inference times in Table A.10.

## 4.2  Benchmark models evaluation

We evaluate the model performance on $test^0$, $test^1_A$ and $test^1_B$. Figure 4 shows *P-R* curves for the different models on label set $test^0$, as well as the curves for all three label sets on the best performing model, *SSD*. Notably, though the model was trained and validated on labels created at $t^0$, the best *P-R* curve is obtained for $test^1_B$, indicating that the trained model is more in agreement with these labels. This suggests that the higher label noise present in $test^0$, as assumed in Section 3.3, was not picked up by the model during training, illustrating once more the resilience of *DL* to label noise (Rolnick et al., 2017). Table 3 presents further evaluation metrics and confirms that *SSD* is the best-performing model overall, while at the standard model confidence threshold of 0.5 *YOLOv3* performs equally well if not better on some label sets. Interestingly, different models exhibit very different Precision-Recall trade-offs at 0.5 model confidence.

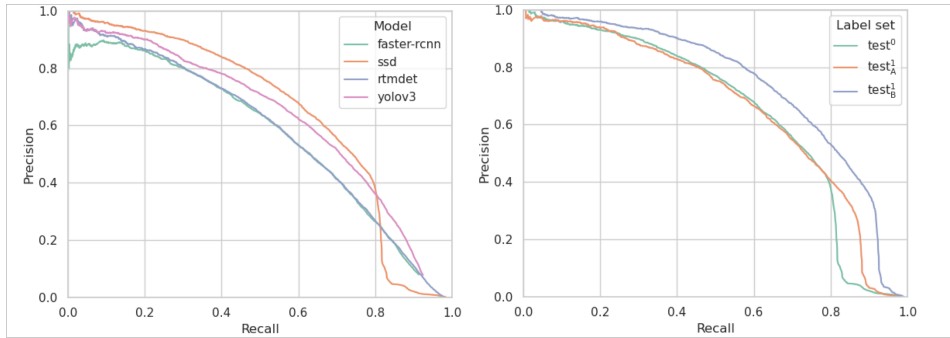

Figure 4: Model Benchmark. *P-R* curves using $test^0$ as the *GT* for all models (left) and using all three label sets for *SSD* (right). We observe that overall the SSD model predictions are more in agreement with the annotations and have a better trade-off between precision and recall. Although the model was trained and validated with labels from $t^0$ it is more in agreement with annotations from timepoint $t^1$.

---

[1]In addition to the tested methods, we initially implemented DETR with standard hyper-parameters for our dataset, but the training was quite unstable, and the performance much worse than other models. We therefore decided not to report the scores in the manuscript. Furthermore, MedSAM (Ma et al., 2024) and Cellpose (Stringer et al., 2021), segmentation-based approaches, did not work well out-of-the box. Since they would also require segmentation labels for fine-tuning to be useful we did not include these methods for benchmarking.

Table 3: Benchmark metrics on the three label sets. Precision, Recall, and F1-score are reported at 0.5 *IOU* threshold and model confidence. *mAP* is reported at 0.5 and 0.75 IoU threshold. The models exhibit different Precision-Recall tradeoffs. The performance of *SSD* is overall better when considering *mAP*, while at the standard model confidence threshold of 0.5 *YOLOv3* performs equally well if not better on some label sets.

| Metric | Label set | Faster R-CNN | SSD | YOLOv3 | RTMDet |
|---|---|---|---|---|---|
| Precision | $test^0$ | 0.23 | 0.61 | **0.73** | 0.64 |
| | $test^1_A$ | 0.16 | 0.44 | 0.58 | 0.54 |
| | $test^1_B$ | 0.18 | 0.50 | 0.67 | 0.56 |
| | **mean** | **0.19** | **0.52** | **0.66** | **0.58** |
| Recall | $test^0$ | 0.84 | 0.67 | 0.48 | 0.51 |
| | $test^1_A$ | 0.92 | 0.78 | 0.62 | 0.69 |
| | $test^1_B$ | **0.97** | 0.83 | 0.67 | 0.68 |
| | **mean** | **0.91** | **0.76** | **0.59** | **0.63** |
| F1-score | $test^0$ | 0.36 | 0.64 | 0.58 | 0.57 |
| | $test^1_A$ | 0.27 | 0.57 | 0.60 | 0.61 |
| | $test^1_B$ | 0.30 | 0.62 | **0.67** | 0.62 |
| | **mean** | **0.31** | **0.61** | **0.62** | **0.60** |
| mAP@0.5IoU (%) | $test^0$ | 56.56 | 64.40 | 62.55 | 57.71 |
| | $test^1_A$ | 57.09 | 65.79 | 61.11 | 63.87 |
| | $test^1_B$ | 68.36 | **73.88** | 70.25 | 63.23 |
| | **mean** | **60.67** | **68.09** | **64.64** | **61.60** |
| mAP@0.75IoU (%) | $test^0$ | 17.48 | 21.81 | 19.15 | 22.56 |
| | $test^1_A$ | 23.53 | 23.42 | 19.13 | 30.13 |
| | $test^1_B$ | **46.98** | **46.48** | 39.01 | 32.85 |
| | **mean** | **29.33** | **30.57** | **25.76** | **28.51** |

## 4.3 Napari plugin

As described in Appendix A.1.3, *MultiOrg* was created using the open-source image analysis tool *Napari* (Ahlers et al., 2019), together with the initial release, *v.0.1.0*, of the *napari-organoid-counter* plugin (Bukas, 2022). In this work, we release a new version of the plugin, *v.0.2.2*, using the model from our benchmark with the better trade-off between performance and inference time as the backbone (see Table A.10), i.e. *YOLOv3*, along with added functionalities. For example, the model confidence threshold can now be adjusted at run time by the user, depending on whether for the task at hand, a higher Recall or Precision is most practical. Plugin details can be found in Appendix A.3. Code and tutorials for installation are distributed through the napari hub

## 5 Discussion

In this work, we release *MultiOrg*, a large multi-rater dataset for organoid detection in 2d microscopy images. Our dataset consists of more than 60,000 annotated lung organoids, labeled by two expert annotators. As even expert annotators can disagree on what constitutes an organoid in those images while also being susceptible to human error and biases, we provide three label sets for the test data, enabling quantification of label uncertainty on a multi- and single-annotator level. Additionally, we have carefully included diversity in our dataset through several study setups and cell lines to ensure good generalization. We performed preliminary tests of the selected model on different lung cell types not present in the dataset, from both human and mouse organoids, and seen that it generalizes quite well. We also tested it successfully on colon organoids and speculate that it can be used for all organoids with similar shape and size. This dataset is, therefore, uniquely situated between the fields of microscopy and uncertainty quantification. We invite researchers to use it, participate in the *MultiOrg* challenge, and assist us in studying label noise in challenging real-life biomedical settings, and we believe that future models should, as much as possible, refrain from being trained without considering these aspects. We additionally publish a benchmark for organoid object detection, provide all models in zenodo and the best one as a Napari plugin, thus enabling scientists potentially use them on their own data.

This work offers a valuable dataset that can be leveraged to advance both object detection methods and uncertainty quantification techniques. The current setting does not, however, permit the incorporation of several label sets in the training loop. Furthermore, it is important to note that although two organoid types are present in the images, they have not been annotated as different classes. Treating this task as a multi-class detection problem may boost the overall performance of the object detection task (Zhang et al., 2022) and would provide added value to the biologists. However, while providing several label sets or multi-class labels on the training data could be beneficial, it represents substantial manual work. It would also be interesting to observe how the label sets change by using *DL* models as a baseline for annotation rather than the pseudo labels. Despite these limitations, we are confident that the release of the *MultiOrg* dataset offers several invaluable contributions to the machine learning community.

## Acknowledgements

This work was funded by the German Center for Lung Research (DZL) and from the Deutsche Forschungsgemeinschaft (DFG, German Research Foundation– 512453064) as well as from the Stiftung Atemweg. We would like to thank Kathrin Federl for her excellent technical assistance during the creation of the dataset. We would also like to thank Isra Mekki and Francesco Campi for reviewing the code for reproducing the benchmark, and Theresa Willem for her help with the Ethics Statement.

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

# A  Appendix

## A.1  Data Description

### A.1.1  Biological study setup

Two different study setups were used to create our dataset. In the first, we used isolated murine distal epithelial cells, enriched for alveolar epithelial type II cells (AEC2), which have the stem-cell function of differentiating into other cell types, hence serving as progenitor cells in the lung (Barkauskas et al., 2013). In addition to the AEC2, we used a murine fibroblast cell line as mesenchymal support cells. Images containing organoids deriving from these biological cultures, are henceforth mentioned as belonging to the 'Normal' setup.

As a second study setup, we added a cell type, namely macrophages, to the organoid culture, which we henceforth refer to as the 'Macros' setup. Isolation of cells and culturing of organoids were performed as previously described in Lehmann et al. (2020). Organoids were seeded as duplicates in 96-well imaging plates with a glass bottom (see Figure 1).

### A.1.2  Image acquisition

The plates used for culturing and imaging were *Falcon® 96-well Black/Clear Flat Bottom TC-treated Imaging Microplates*. The brightfield images were acquired with a Life Cell Imaging Microscope *(LifeCellImagerObserver.Z1)* at a 5x objective. During the acquisition of the images, each well was divided into tiles and stacks to capture the 3d growth of organoids. Per well, 24 tiles and 10-15 stacks were acquired. Individual tiles were stitched together to form one single image per plate. We observed that most object detection methods can successfully detect organoids on the borders of two or more patches, even if the stitching mechanism is imperfect. Therefore, in our setup, we decided to work with the stitched images rather than the individual patches.

Maximum projections were generated by the *Zen 2 Blue* software by *Carl Zeiss Microscopy GmbH* to process the image stacks into one plain 2D image, such that each pixel in the final image derives from the slice in the stack which is most in focus at that location. Since organoid structures are relatively spherical, one can easily approximate their area using a 2D projection. Additionally, labeling images in 2D greatly speeds up the annotation procedure. Images were exported in the *CZI* file format.

The resulting 2D images have varying sizes, between 5719 and 6240 pixels in the *x* and 5551 and 6940 pixels in the *y* axis, respectively. Each pixel in the image is equivalent to 1.29 $\mu m$ in each axis. At this point, the images were examined, and eight plates were dropped, either due to lower image quality or because the organoid formation did not work well, resulting in noisy images. The latter was mainly observed in the Macros study setup, which resulted in fewer data from this setup in our final dataset. Finally, the imaged wells were randomly selected by plates and study setups to be annotated using our annotation tool of choice (see Appendix A.1.3).

### A.1.3  Annotation Procedure

The annotation process was carried out by running the initial release, *v.0.1.0*, of the *napari-organoid-counter* tool (Bukas, 2022), a plugin developed for *Napari* (Ahlers et al., 2019). The tool parameters were set to a down-sampling of one, minimal diameter of 30 µm and sigma of three. After running the *napari-organoid-counter* with these parameters, all detected organoids were examined. All spherical structures consisting of visibly more than one cell and measuring more than 30 µm, were recognized as organoids. The wrongfully created box was manually deleted if the counter detected a *False Positive (FP)*. If the counter detected the organoid size or exact location incorrectly, the box was manually moved or adjusted according to the correct size and location. If the counter detected accumulations of organoids as one single object, the box was deleted and correctly sized boxes for the single organoids were created. If the counter did not detect an organoid, a box according to the organoid's size was manually created.

### A.1.4  Data preparation for release

After all the data was collected and annotated, all images were converted from the proprietary CZI to the open TIFF format. Additionally, all studies were renamed to ensure consistency and all images and annotation information for each image were anonymized.

Table A.4: A detailed overview of the dataset. The training set consists of 356 images derived from 25 studies, and the test set consists of 55 images from 7 studies.

| Study Setup | Plate Name | Number of Wells | Image IDs | Data split | Annotator |
|---|---|---|---|---|---|
| Normal | Plate_11 | 13 | 1-13 | train | A |
| Normal | Plate_13 | 1 | 14 | train | A |
| Macros | Plate_13 | 6 | 15-20 | train | A |
| Normal | Plate_19 | 6 | 21-26 | train | A |
| Normal | Plate_20 | 20 | 27-46 | train | A |
| Normal | Plate_26 | 34 | 47-80 | train | A |
| Normal | Plate_29 | 26 | 81-106 | train | A |
| Normal | Plate_3 | 5 | 107-111 | train | A |
| Normal | Plate_32 | 19 | 112-130 | train | A |
| Normal | Plate_33 | 16 | 131-146 | train | A |
| Normal | Plate_34 | 16 | 147-162 | train | A |
| Macros | Plate_6 | 9 | 163-171 | train | A |
| Normal | Plate_8 | 10 | 172-181 | train | A |
| Normal | Plate_9 | 15 | 182-196 | train | A |
| Normal | Plate_16 | 17 | 197-213 | train | B |
| Macros | Plate_16 | 9 | 214-222 | train | B |
| Normal | Plate_17 | 18 | 223-240 | train | B |
| Macros | Plate_17 | 4 | 241-244 | train | B |
| Normal | Plate_18 | 34 | 245-278 | train | B |
| Macros | Plate_18 | 6 | 279-284 | train | B |
| Normal | Plate_24 | 10 | 285-294 | train | B |
| Macros | Plate_25 | 6 | 295-300 | train | B |
| Normal | Plate_36 | 15 | 301-315 | train | B |
| Normal | Plate_39 | 17 | 316-332 | train | B |
| Normal | Plate_40 | 24 | 333-356 | train | B |
| Normal | Plate_37 | 6 | 1-6 | test | A |
| Normal | Plate_4 | 2 | 7-8 | test | A |
| Macros | Plate_4 | 14 | 9-22 | test | A |
| Normal | Plate_15 | 12 | 23-34 | test | B |
| Normal | Plate_31 | 8 | 35-42 | test | B |
| Macros | Plate_15 | 7 | 43-49 | test | B |
| Macros | Plate_23 | 6 | 50-55 | test | B |

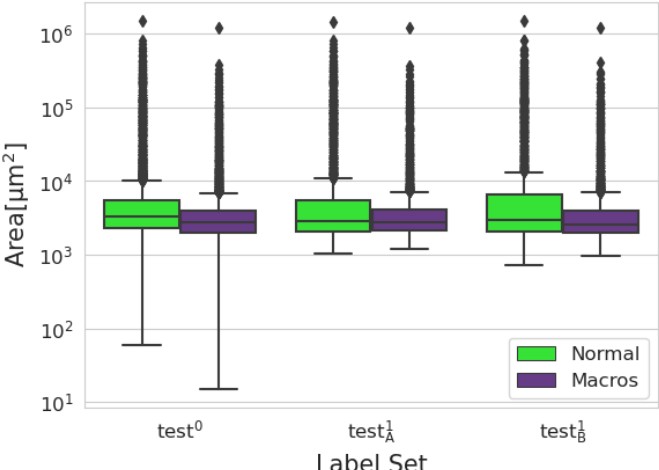

Figure A.5: Bounding box sizes. Box plots of the bounding box areas in $test^0$, $test^1_A$ and $test^1_B$, stratified by study type, on a logarithmic scale.

Table A.5: Bounding box sizes. Mean and standard deviation of the bounding box areas (in $\mu m^2$) in $test_A^0$, $test_B^0$, $test_A^1$, and $test_B^1$, stratified by study type and combined.

| Label Subset / Study Type | Normal | Macros | Combined |
|---|---|---|---|
| $test_A^0$ | 11,887($\pm$ 51,000) | 4,891($\pm$ 31,918) | 7,553($\pm$ 40,400) |
| $test_B^0$ | 17,064($\pm$ 57,976) | 9,654($\pm$ 25,272) | 14,612($\pm$ 49,726) |
| $test_A^1$ | 19,413($\pm$ 64,855) | 8,714($\pm$ 36,128) | 14,931($\pm$ 14,931) |
| $test_B^1$ | 21,315($\pm$ 68,956) | 8,188($\pm$ 35,042) | 15,220($\pm$ 56,215) |

### A.1.5 Multi-rater analysis

Table A.6: Statistics of the multi-rater analysis. The intra-rater scores for Annotator A are calculated for $test_A^0$ (considered as *GT*) and the corresponding subset of $test_A^1$ for images 1-22. Similarly, for Annotator B, the intra-rater score is computed between $test_B^0$ (considered as *GT*) and the corresponding subset of $test_B^1$ for images 23-55. Inter-rater scores for images 1-22 are calculated between $test_A^0$ (considered as *GT*) and the corresponding subset of $test_B^1$, as well as subsets of $test_A^1$ (considered as *GT*) and $test_B^1$. Similarly, for images 23-55, the inter-rater scores are computed between $test_B^0$ (considered as *GT*) and the corresponding subset of $test_A^1$, as well as subsets of $test_A^1$ (considered as *GT*) and $test_B^1$.

| | Images 1-22 | | | | | | | | |
|---|---|---|---|---|---|---|---|---|---|
| | **Intra-rater Annotator A** | | | **Inter-rater $test_A^0$ vs. $test_B^1$** | | | **Inter-rater $test_A^1$ vs. $test_B^1$** | | |
| | F1-Score | Precision | Recall | F1-Score | Precision | Recall | F1-Score | Precision | Recall |
| **Median** | 0.624 | 0.806 | 0.516 | 0.586 | 0.813 | 0.469 | 0.610 | 0.689 | 0.604 |
| **Mean** | 0.635 | 0.787 | 0.551 | 0.577 | 0.782 | 0.473 | 0.605 | 0.677 | 0.595 |
| **Std** | 0.134 | 0.094 | 0.173 | 0.090 | 0.091 | 0.116 | 0.110 | 0.175 | 0.159 |
| **min** | 0.305 | 0.581 | 0.207 | 0.397 | 0.569 | 0.258 | 0.360 | 0.276 | 0.320 |
| **25%** | 0.545 | 0.726 | 0.439 | 0.523 | 0.731 | 0.414 | 0.547 | 0.593 | 0.481 |
| **50%** | 0.624 | 0.801 | 0.516 | 0.586 | 0.813 | 0.470 | 0.610 | 0.689 | 0.604 |
| **75%** | 0.766 | 0.848 | 0.740 | 0.649 | 0.854 | 0.547 | 0.690 | 0.822 | 0.734 |
| **max** | 0.826 | 0.809 | 0.939 | 0.754 | 0.889 | 0.692 | 0.767 | 0.922 | 0.848 |

| | Images 23-55 | | | | | | | | |
|---|---|---|---|---|---|---|---|---|---|
| | **Intra-rater Annotator B** | | | **Inter-rater $test_B^0$ vs. $test_A^1$** | | | **Inter-rater $test_A^1$ vs. $test_B^1$** | | |
| | F1-Score | Precision | Recall | F1-Score | Precision | Recall | F1-Score | Precision | Recall |
| **Median** | 0.643 | 0.800 | 0.546 | 0.561 | 0.725 | 0.426 | 0.710 | 0.667 | 0.761 |
| **Mean** | 0.632 | 0.778 | 0.551 | 0.522 | 0.752 | 0.414 | 0.655 | 0.608 | 0.761 |
| **Std** | 0.127 | 0.136 | 0.155 | 0.140 | 0.159 | 0.145 | 0.145 | 0.188 | 0.133 |
| **min** | 0.193 | 0.282 | 0.147 | 0.219 | 0.348 | 0.148 | 0.281 | 0.222 | 0.381 |
| **25%** | 0.568 | 0.729 | 0.461 | 0.434 | 0.650 | 0.324 | 0.559 | 0.480 | 0.704 |
| **50%** | 0.643 | 0.800 | 0.546 | 0.561 | 0.725 | 0.426 | 0.710 | 0.667 | 0.761 |
| **75%** | 0.703 | 0.885 | 0.667 | 0.611 | 0.909 | 0.503 | 0.760 | 0.750 | 0.835 |
| **max** | 0.844 | 0.940 | 0.878 | 0.807 | 1.000 | 0.732 | 0.828 | 0.831 | 0.988 |

Table A.7: Statistics of the multi-rater analysis for 'Normal' images. The intra-rater scores for Annotator A are calculated for $test_A^0$ (considered as *GT*) and the corresponding subset of $test_A^1$ for images 9-22. Similarly, for Annotator B, the intra-rater score is computed between $test_B^0$ (considered as *GT*) and the corresponding subset of $test_B^1$ for images 43-55. Inter-rater scores for images 9-22 are calculated between $test_A^0$ (considered as *GT*) and the corresponding subset of $test_B^1$, as well as subsets of $test_A^1$ (considered as *GT*) and $test_B^1$. Similarly, for images 43-55, the inter-rater scores are computed between $test_B^0$ (considered as *GT*) and the corresponding subset of $test_A^1$, as well as subsets of $test_A^1$ (considered as *GT*) and $test_B^1$.

| | Images 1-8 | | | | | | | | |
|---|---|---|---|---|---|---|---|---|---|
| | **Intra-rater Annotator A** | | | **Inter-rater $test_A^0$ vs. $test_B^1$** | | | **Inter-rater $test_A^1$ vs. $test_B^1$** | | |
| | F1-Score | Precision | Recall | F1-Score | Precision | Recall | F1-Score | Precision | Recall |
| **Median** | 0.804 | 0.845 | 0.779 | 0.584 | 0.843 | 0.444 | 0.610 | 0.843 | 0.475 |
| **Mean** | 0.768 | 0.824 | 0.735 | 0.584 | 0.848 | 0.458 | 0.612 | 0.834 | 0.513 |
| **Std** | 0.081 | 0.090 | 0.117 | 0.100 | 0.028 | 0.127 | 0.096 | 0.082 | 0.171 |
| **min** | 0.617 | 0.629 | 0.459 | 0.454 | 0.814 | 0.308 | 0.475 | 0.651 | 0.320 |
| **25%** | 0.761 | 0.826 | 0.740 | 0.532 | 0.827 | 0.382 | 0.558 | 0.822 | 0.410 |
| **50%** | 0.804 | 0.845 | 0.779 | 0.584 | 0.843 | 0.444 | 0.610 | 0.843 | 0.475 |
| **75%** | 0.817 | 0.855 | 0.799 | 0.638 | 0.867 | 0.524 | 0.696 | 0.882 | 0.600 |
| **max** | 0.826 | 0.939 | 0.809 | 0.754 | 0.889 | 0.692 | 0.737 | 0.922 | 0.848 |
| | **Images 23-42** | | | | | | | | |
| | **Intra-rater Annotator B** | | | **Inter-rater $test_B^0$ vs. $test_A^1$** | | | **Inter-rater $test_A^1$ vs. $test_B^1$** | | |
| | F1-Score | Precision | Recall | F1-Score | Precision | Recall | F1-Score | Precision | Recall |
| **Median** | 0.657 | 0.794 | 0.564 | 0.578 | 0.833 | 0.444 | 0.738 | 0.707 | 0.823 |
| **Mean** | 0.645 | 0.784 | 0.564 | 0.559 | 0.819 | 0.435 | 0.696 | 0.632 | 0.823 |
| **Std** | 0.155 | 0.150 | 0.179 | 0.139 | 0.161 | 0.134 | 0.117 | 0.166 | 0.104 |
| **min** | 0.193 | 0.282 | 0.147 | 0.219 | 0.348 | 0.160 | 0.404 | 0.268 | 0.595 |
| **25%** | 0.593 | 0.753 | 0.469 | 0.486 | 0.744 | 0.356 | 0.639 | 0.532 | 0.758 |
| **50%** | 0.657 | 0.794 | 0.564 | 0.578 | 0.833 | 0.444 | 0.738 | 0.707 | 0.823 |
| **75%** | 0.719 | 0.891 | 0.679 | 0.636 | 0.932 | 0.504 | 0.779 | 0.750 | 0.891 |
| **max** | 0.844 | 0.940 | 0.878 | 0.807 | 1.000 | 0.684 | 0.828 | 0.822 | 0.988 |

Table A.8: Statistics of the multi-rater analysis for 'Macros' images. The intra-rater scores for Annotator A are calculated for $test_A^0$ (considered as *GT*) and the corresponding subset of $test_A^1$ for images 9-22. Similarly, for Annotator B, the intra-rater score is computed between $test_B^0$ (considered as *GT*) and the corresponding subset of $test_B^1$ for images 43-55. Inter-rater scores for images 9-22 are calculated between $test_A^0$ (considered as *GT*) and the corresponding subset of $test_B^1$, as well as subsets of $test_A^1$ (considered as *GT*) and $test_B^1$. Similarly, for images 43-55, the inter-rater scores are computed between $test_B^0$ (considered as *GT*) and the corresponding subset of $test_A^1$, as well as subsets of $test_A^1$ (considered as *GT*) and $test_B^1$.

| | Images 9-22 | | | | | | | | |
|---|---|---|---|---|---|---|---|---|---|
| | **Intra-rater Annotator A** | | | **Inter-rater $test_A^0$ vs. $test_B^1$** | | | **Inter-rater $test_A^1$ vs. $test_B^1$** | | |
| | F1-Score | Precision | Recall | F1-Score | Precision | Recall | F1-Score | Precision | Recall |
| **Median** | 0.556 | 0.769 | 0.449 | 0.586 | 0.744 | 0.486 | 0.621 | 0.615 | 0.682 |
| **Mean** | 0.560 | 0.766 | 0.446 | 0.573 | 0.744 | 0.482 | 0.600 | 0.587 | 0.643 |
| **Std** | 0.091 | 0.092 | 0.090 | 0.087 | 0.093 | 0.113 | 0.120 | 0.148 | 0.136 |
| **min** | 0.305 | 0.581 | 0.207 | 0.397 | 0.569 | 0.258 | 0.360 | 0.276 | 0.370 |
| **25%** | 0.539 | 0.712 | 0.411 | 0.522 | 0.667 | 0.439 | 0.546 | 0.512 | 0.517 |
| **50%** | 0.556 | 0.769 | 0.449 | 0.586 | 0.744 | 0.486 | 0.621 | 0.615 | 0.682 |
| **75%** | 0.620 | 0.832 | 0.524 | 0.646 | 0.812 | 0.557 | 0.688 | 0.698 | 0.748 |
| **max** | 0.683 | 0.935 | 0.542 | 0.684 | 0.871 | 0.640 | 0.767 | 0.771 | 0.804 |
| | **Images 43-55** | | | | | | | | |
| | **Intra-rater Annotator B** | | | **Inter-rater $test_B^0$ vs. $test_A^1$** | | | **Inter-rater $test_A^1$ vs. $test_B^1$** | | |
| | F1-Score | Precision | Recall | F1-Score | Precision | Recall | F1-Score | Precision | Recall |
| **Median** | 0.626 | 0.806 | 0.503 | 0.491 | 0.659 | 0.373 | 0.648 | 0.655 | 0.704 |
| **Mean** | 0.613 | 0.768 | 0.529 | 0.464 | 0.650 | 0.381 | 0.592 | 0.569 | 0.665 |
| **Std** | 0.063 | 0.117 | 0.111 | 0.127 | 0.087 | 0.160 | 0.164 | 0.219 | 0.116 |
| **min** | 0.515 | 0.590 | 0.368 | 0.226 | 0.476 | 0.148 | 0.281 | 0.222 | 0.116 |
| **25%** | 0.560 | 0.647 | 0.461 | 0.405 | 0.586 | 0.296 | 0.523 | 0.433 | 0.636 |
| **50%** | 0.626 | 0.806 | 0.503 | 0.491 | 0.659 | 0.373 | 0.648 | 0.655 | 0.704 |
| **75%** | 0.662 | 0.862 | 0.606 | 0.561 | 0.705 | 0.435 | 0.727 | 0.700 | 0.752 |
| **max** | 0.723 | 0.905 | 0.738 | 0.651 | 0.790 | 0.732 | 0.776 | 0.831 | 0.778 |

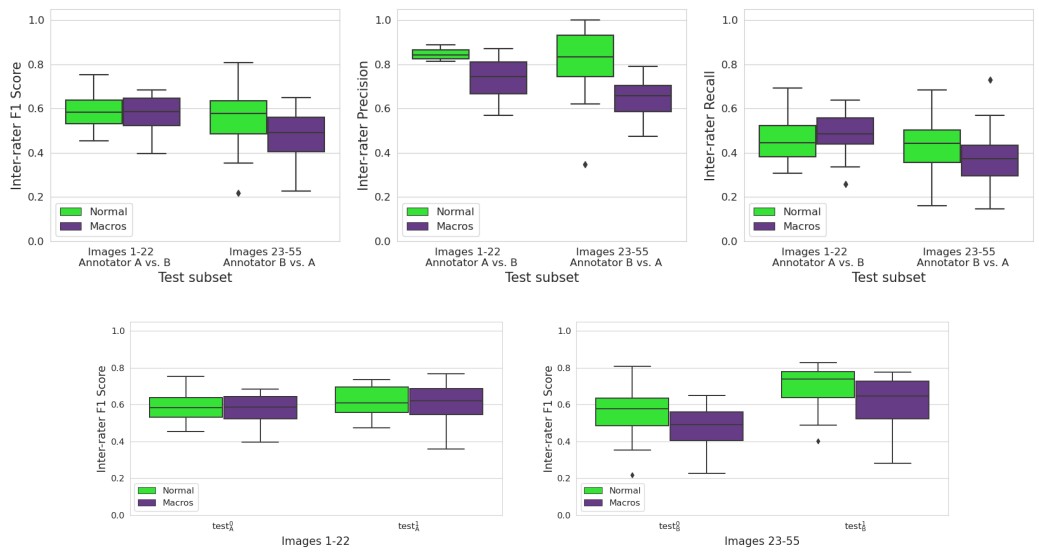

Figure A.6: Inter-rater scores for the two subsets of the test set. **Top**: F1-score (left), Precision (middle), and Recall (right) shown for Intra- and Inter-rater scores for both annotators. Scores are split for the two test subsets and according to study type. $test^0$ is always considered the *GT* for computing these scores. **Bottom**: F1-score on images 1-22 (left), where $test_B^1$ is used as the *GT* and scores are computed against $test_A^0$ and $test_A^1$, and on images 23-55 (right), where $test_A^1$ is used as the *GT* and scores are computed against $test_B^0$ and $test_B^1$. On the left, we see that when compared to a third independent label set corresponding to a different annotator, annotator A has slightly changed their style of annotation at timepoint $t^1$, slightly converging annotator B. On the right, we see an even bigger shift in annotation style. These results suggest that annotators exchanged best practices in annotation styles between timepoints $t^0$ and $t^1$.

Table A.9: Comparison of the label sets to pseudo labels. Precision, Recall, and F1-score computed between the pseudo labels and $test^0$, $test_A^1$, $test_B^1$ (considered as *GT* in this computation). The pseudo labels are used as a starting point by each annotator to annotate the data. The F1-score shows that $test_B^1$ was curated the least, while the higher Recall values at time point $t^1$ confirm that more of the pseudo labels were removed than at $t^0$.

| Metric | $test^0$ | $test_A^1$ | $test_B^1$ |
|---|---|---|---|
| Precision | 0.51 | 0.40 | 0.48 |
| Recall | 0.23 | 0.28 | 0.33 |
| F1-score | 0.32 | 0.33 | 0.39 |

### A.2 Model benchmark

### A.2.1 Training details

For the training and validation pipeline of our benchmark the *mmdetection* (Chen et al., 2019) toolbox was used, a well-established open-source toolbox for object detection. To make our dataset compatible with the toolbox, the bounding boxes were converted to the COCO format (Lin et al., 2015). We adapted the original configuration for each model to set a number of fixed parameters for all models. AdamW was used as the optimizer with a base learning rate of 1e-05, along with a linear learning rate scheduler. The batch size was set to 16 and the training setup included standard image augmentations: Gaussian Blur, Random Flip, Random Shift, Random Affine, and Photometric Distortion with a probability of 0.5. For all models, the pretrained COCO weights were used as initialization and the final layer was adapted to accommodate our single class. All models were trained for 400 epochs and validated using the COCO metrics on $test^0$. Training and validation were performed on an internal cluster that uses an NVIDIA A100 GPU with four cores and 40 GB VRAM. The training time varied slightly depending on the model, but all were trained in less than 21 hours (Table A.10).

Table A.10: Training and testing of benchmark models. Train time is the duration of training the model once on the entire train set for 400 epochs. The best epoch is the epoch with the highest mAP on the test set $test^0$. GPU utilization indicates the approximate range of GPU utilization during training. Inference time is the average time per image inference using a single core.

| Model | Faster R-CNN | SSD | YOLOv3 | RTMDet |
|---|---|---|---|---|
| Train time (hours) | 15 | 16 | 10 | 20 |
| Best epoch | 68 | 86 | 27 | 323 |
| Model size (MB) | 172 | 99 | 249 | 454 |
| GPU utilization (%) | 60-70 | 30-90 | 55-65 | 40-80 |
| Inference time (seconds) | 18 | 59 | 13 | 114 |

### A.2.2 Additional results

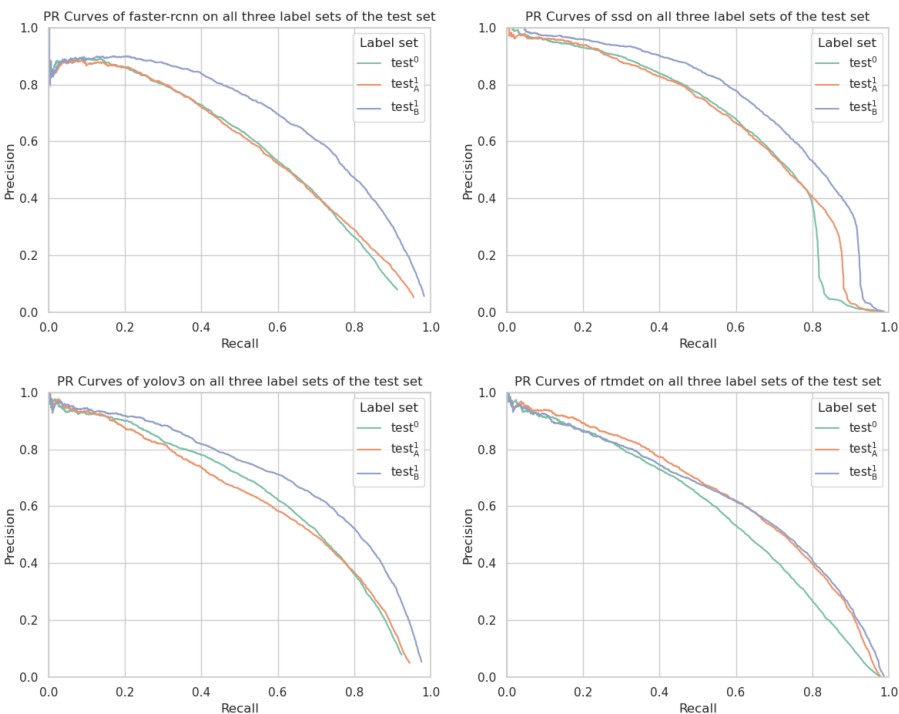

Figure A.7: Model benchmark. *P-R* curves across all models of the benchmark for all three label sets. It is interesting to observe that though the model checkpoints were selected based on $test^0$, they are consistently more in agreement with labels of $test_B^1$.

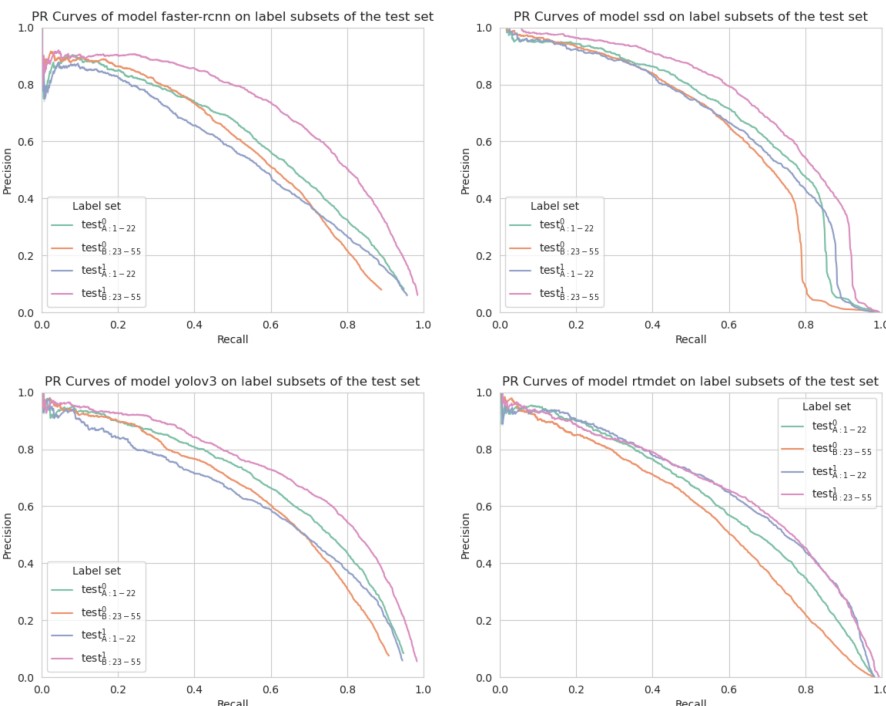

Figure A.8: Model benchmark. *P-R* curves across all models of the benchmark for subsets of the label sets: $test^0_{A:1-22}$ and $test^0_{B:23-55}$, along with the corresponding subsets for $test^1_A$ and $test^1_B$, i.e. $test^1_{A:1-22}$ and $test^1_{B:23-55}$ such that a direct comparison of the same subsets of the test set can be made. We see that all models are more in agreement with Annotator B at timepoint $t^1$ compared to timepoint $t^0$ for images 23-55 of the test set.

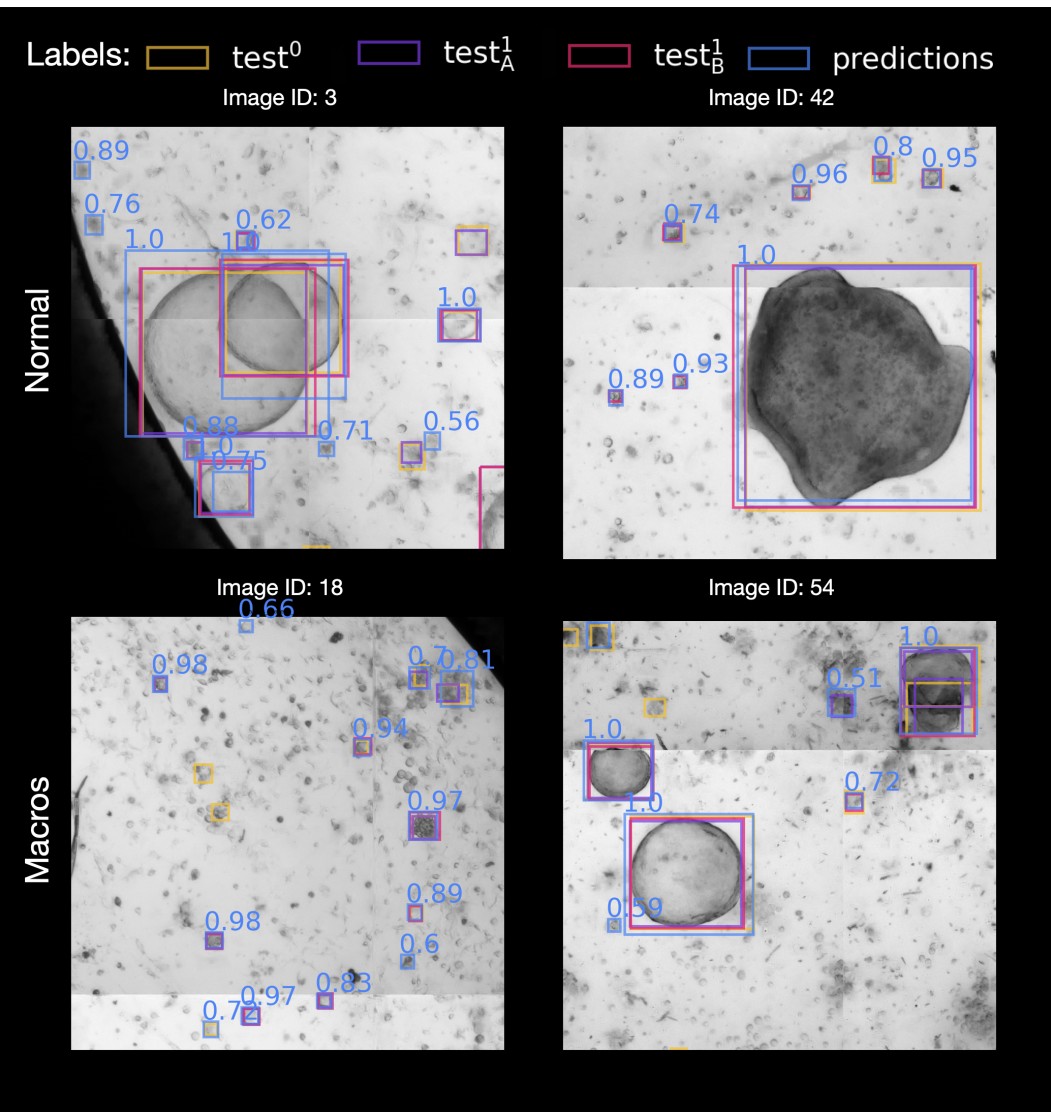

Figure A.9: Example Predictions. Predicted Bounding boxes and model confidence from the *SSD* model for various image crops of size 1000x1000 pixels of the 'Normal' (top) and 'Macros' (bottom) study types. The three label sets are also displayed for comparison. The 'Macros' images are noisier and, therefore, more challenging for the model and the annotators alike.

## A.3 Extensions of napari-organoid- counter v.0.2.

The main extensions of the latest version of the *napari-organoid-counter* plugin are as follows:

- Back-end: Use of the trained *YOLOv3* model presented in Section 4.1 for object detection. This model was chosen as it gives the best trade-off between performance and inference time for real-time applications.
- Back-end: Inference with a sliding window and adjustable parameters for multiple window sizes and window down-sampling rates.
- Back-end: Adjustable model confidence threshold.
- Front-end: Organoid ID and model confidence displayed in the viewer - the individual exported features can now be traced back to the organoids in the viewer.
- Front-end: Possibility to work interactively with different layers simultaneously by adjusting parameters and switching between shape layers.

## A.4 Ethics Statement

We have thoroughly reviewed this work for any potential ethical implications and believe that the societal benefits outweigh the potential issues related to this work.

The dataset we are submitting consists of lung organoids derived from murine cells. The data we provide is original, and any supplementary data included adheres to the Creative Commons licensing terms. We have verified that all datasets used in our submission are current and have not been deprecated by their original creators. Privacy-related concerns are minimal, as our dataset does not involve human subjects and, therefore, does not entail sensitive personal data, or sensitive information of humans, thereby mitigating common ethical concerns related to privacy and consent.

The dataset is intended for research within the scope of lung organoids derived from murine cells, and its applicability is limited to this specific area. Users of the dataset are encouraged to consider this limitation and ensure their research appropriately reflects the dataset's scope and intended use when taking up our work and using it beyond the specific biological domain for which we created the dataset. Prior to using this dataset the intended use of the model should be considered against the scope and, if the dataset is used outside the scope we refer to, it should be thoroughly tested regarding the new context to avoid biases and ensure the reliability of the resulting models.

No matter the context the dataset is used in, we would like to emphasize the importance of maintaining a human in the loop for final decision-making processes to avoid over-reliance on automated systems.

However, we nevertheless emphasize the importance of responsible use of this dataset. Researchers utilizing this dataset should ensure that all analyses and applications are conducted within ethical guidelines and that any machine learning models or automated tools developed using this data are implemented with care.

# B MultiOrg Datasheet

This data sheet serves as supplementary documentation aimed at improving reproducibility. It is based on the guidelines outlined in **Datasheets for Datasets**[2], a working paper developed for the machine learning community.

- **For what purpose was the dataset created?** This dataset was created with two goals in mind: a. to facilitate research in uncertainty quantification methods in machine learning and b. to enable the development of object detection models for the automated detection of lung organoids, which can accelerate the annotation process for similar data in future studies.

- **Who created the dataset (e.g., which team, research group) and on behalf of which entity (e.g., company, institution, organization)?** The dataset was created as a collaborative effort by the authors of this work, i.e. biologists of the Institute of Lung Health and Immunity (LHI) of the Helmholtz Zentrum München and the Philipp University of Marburg, as well as computer scientists Helmholtz AI at the Helmholtz Zentrum München.

- **Who funded the creation of the dataset?** This work was partly funded by the German Center for Lung Research (DZL) and from the Deutsche Forschungsgemeinschaft (DFG, German Research Foundation– 512453064) as well as from the Stiftung Atemweg. Additionally, it was developed as part of the daily work of the creators and was indirectly funded via their salaries.

## B.1 Composition

- **What do the instances that comprise the dataset represent?** Our dataset consists of 2D images of microscopy plate wells, consisting of lung organoids derived from murine cells. Along with the imaging data, annotations for each image are provided, in the form of bounding boxes fit around the organoids, and metadata information on the annotator and time point of annotation.

- **How many instances are there in total (of each type, if appropriate)?** In total 411 fully annotated images are released as part of this dataset, annotated by two annotators at two different timepoints and consisting of 26 different experiments across two biological study setups. Please see Section 3 and Table A.4 for more details and stratification between study types, annotators, train and test splits.

- **Does the dataset contain all possible instances or is it a sample of instances from a larger set?** The dataset is a subset of a larger set of biological experiments. During the first curation of the data, the 26 experiments were selected to have the best representation of our data, with the least noise stemming from the image acquisition and with an acceptable number of organoids in the well, neither too many, which would make it hard to distinguish them from one another, nor too few, with little information present in the image.

- **What data does each instance consist of?** Each instance is an image of size between 5719 and 6240 pixels in the $x$ and 5551 and 6940 pixels in the $y$ axis respectively. Each pixel in the image is equivalent to 1.29 $\mu m$ in each axis.

- **Is there a label or target associated with each instance?** Yes, specifically for the training set one target per image is available, while for the images of the test set, we release three sets of labels. The first, which in this work is named $test^0$, is directly available, while the other two can be indirectly accessed by participating in our Kaggle competition and submitting results to our leaderboard.

- **Is any information missing from individual instances?** No, to the best of our knowledge, all available information has been provided.

- **Are relationships between individual instances made explicit?** Yes, Table A.4 and the structure of the available data make relationships explicit. Relationships become apparent with the data structure provided in the metadata file. For example, all image wells belonging to the experiment "Plate_1" can be found in a folder named "Plate_1".

---

[2] https://arxiv.org/abs/1803.09010

- **Are there recommended data splits (e.g., training, development/validation, testing)?** Yes, we provided the data already split into train and test sets. As discussed in the main manuscript, for validation we use the test data with one of the three label sets.

- **Are there any errors, sources of noise, or redundancies in the dataset?** Noise is always present in microscopy data, and is one of the reasons for which machine learning tasks in the biomedical domain are much harder compared to natural images. This noise can derive from the microscope itself, the imaging parameters, or the biological specimen. Nevertheless, we tried to eliminate these as much as possible when curating the dataset, by, as mentioned above, selecting our experiments out of a larger pool and Another source of noise in our case is the stitching of the imaging tiles to form the 2D image well, which we discuss in Section 3. No redundancies are present in the dataset.

- **Is the dataset self-contained, or does it link to or otherwise rely on external resources (e.g., websites, tweets, other datasets)?** The provided dataset is self-contained.

- **Does the dataset contain data that might be considered confidential (e.g., data that is protected by legal privilege or by doctor-patient confidentiality, data that includes the content of individuals' non-public communications)?** No.

- **Does the dataset contain data that, if viewed directly, might be offensive, insulting, threatening, or might otherwise cause anxiety?** No.

- **Does the dataset relate to people?** No.

## B.2 Collection process

- **How was the data associated with each instance acquired?** The data was acquired by cell culture and imaging in a Life Cell Imaging Microscope.

- **What mechanisms or procedures were used to collect the data?** The biological experiments consisted of isolating and culturing murine cells, which received various treatments and formed variable organoids in the process. Subsequently, images were taken to document and analyze the effects of different treatments.

- **If the dataset is a sample from a larger set, what was the sampling strategy?** The dataset is a subset of a larger set of biological experiments. During the first curation of the data, the 26 experiments were selected to have the best representation of our data, with the least noise stemming from the image acquisition and with an acceptable number of organoids in the well, neither too many, which would make it hard to distinguish them from one another, nor too few, with little information present in the image.

- **Who was involved in the data collection process (e.g., students, crowdworkers, contractors) and how were they compensated (e.g., how much were crowdworkers paid)?** The data was collected by university students with guest contracts and employees of the Helmholtz Zentrum Munich.

- **Over what timeframe was the data collected?** The data was collected at specific timepoints over a period of two weeks. All experiments took place over a timeframe of two years.

- **Were any ethical review processes conducted (e.g., by an institutional review board)?** No

- **Does the dataset relate to people?** No, it is a murine dataset.

## B.3 Preprocessing/cleaning/labeling

- **Was any preprocessing/cleaning/labeling of the data done ?** The data was indeed preprocessed, cleaned, and labeled. In the Appendix, we describe these processes under Appendix A.1.3 and Appendix A.1.2.

- **Was the 'raw' data saved in addition to the preprocessed/cleaned/labeled data (e.g., to support unanticipated future uses)?** Yes, the raw data was saved and stored.

- **Is the software used to preprocess/clean/label the instances available?** The software used for preprocessing the data, *Zen 2 Blue* software by *Carl Zeiss Microscopy GmbH*, is proprietary. The software used for annotating the data of the *napari-organoid-counter* tool (Bukas, 2022), a plugin developed for *Napari*, is open-source and freely available.

### B.4 Uses

- **Has the dataset been used for any tasks already?** The dataset was used as part of this work to create the benchmark presented in Section 4. The trained models are also made publicly available through this work and can be accessed on zenodo. Moreover, the latest version of the *napari-organoid-counter* tool described in Section 4.3 also uses one of the models from this benchmark, trained with the current dataset.

- **Is there a repository that links to any or all papers or systems that use the dataset?** If so, please provide a link or other access point.

  No, please refer to Appendix C instead.

- **What (other) tasks could the dataset be used for?** We discussed previously how the intended usage of our dataset is two-fold, a. to facilitate research in uncertainty quantification methods in machine learning and b. to enable the development of object detection models for the automated detection of lung organoids, which can accelerate the annotation process for similar data in future studies. Aside from these tasks, one could use this dataset to benchmark new model architectures for object detection, or to develop unsupervised learning methods for organoid classification (using the bounding boxes to extract single organoid images), since we mention that our dataset consists of two different types of organoids.

- **Is there anything about the composition of the dataset or the way it was collected and preprocessed/cleaned/labeled that might impact future uses?** No.

- **Are there tasks for which the dataset should not be used?** Not to the best of our knowledge.

### B.5 Distribution

- **Will the dataset be distributed to third parties outside of the entity (e.g., company, institution, organization) on behalf of which the dataset was created?** Yes, the dataset is hereby made publicly available under the CC BY-NC-SA 4.0 License, and can therefore be used by third parties.

- **How will the dataset will be distributed (e.g., tarball on the website, API, GitHub)?** The dataset is hereby made available via the Kaggle platform and can be accessed through the link: https://www.kaggle.com/datasets/christinabukas/mutliorg/ and DOI: 10.34740/kaggle/ds/5097172

- **When will the dataset be distributed?** The dataset is made available along with the submission of the current manuscript.

- **Will the dataset be distributed under a copyright or other intellectual property (IP) license, and/or under applicable terms of use (ToU)?** Our dataset is open source and made available under the CC BY-NC-SA 4.0 License.

- **Have any third parties imposed IP-based or other restrictions on the data associated with the instances?** No.

- **Do any export controls or other regulatory restrictions apply to the dataset or to individual instances?** No.

### B.6 Maintenance

- **Who is supporting/hosting/maintaining the dataset?** The dataset is maintained by the authors of this work. The dataset is currently hosted on the Kaggle[3] platform.

- **How can the owner/curator/manager of the dataset be contacted (e.g., email address)?** Kaggle offers a discussion tab under the dataset repository. This can be used for any data-related discussions, while there is also a discussion tab available on the website of our competition. Naturally, the corresponding authors of this work may also be contacted directly with any questions via email.

- **Is there an erratum?** There is currently no erratum for this dataset.

---

[3] https://www.kaggle.com

- **Will the dataset be updated (e.g., to correct labeling errors, add new instances, delete instances)?** There are currently no immediate plans for updating the dataset. We are eager to first see how it will be accepted by the community and which needs will arise for future versions/extensions. We, of course, plan to maintain the dataset, e.g. if errors are found in the data, we will update the dataset and release a newer version.

- **If the dataset relates to people, are there applicable limits on the retention of the data associated with the instances?** The dataset does not relate to people.

- **Will older versions of the dataset continue to be supported/hosted/maintained?** If and when newer versions of the dataset are released we expect these to be an improvement upon the original version, and will therefore concentrate our efforts on maintaining the latest version of the dataset.

- **If others want to extend/augment/build on/contribute to the dataset, is there a mechanism for them to do so?** Researchers are more than welcome to extend our dataset. In the discussion section of our manuscript, Section 5, we mention how future versions of our dataset could include even more label sets both for the train and test sets. Such versions would surely increase its value and the development of uncertainty quantification techniques.

## C  Availability: data, benchmark, and software tool

Below we list all assets made publicly available with the release of this work:

- Trained model weights from our benchmark can be found on zenodo with a DOI: 10.5281/zenodo.11258022
- The MultiOrg dataset can be found on Kaggle with a DOI: 10.34740/kaggle/ds/5097172
- The notebooks for reproducing our benchmark can be found under the same repository on the Kaggle MultiOrg dataset page
- The Croissant metadata record documenting the dataset can also be found on the Kaggle MultiOrg dataset page here
- The napari plugin can be found on the napari-hub

# D  Author Statement

As the authors of this dataset, we hereby declare that we bear full responsibility for any and all consequences arising from the use, distribution, and publication of the dataset. This includes but is not limited to, any violations of privacy, intellectual property rights, or any other legal rights.

We confirm that all data included in this dataset has been collected, processed, and shared in compliance with applicable laws and regulations. We have obtained all necessary permissions and consents from individuals or entities involved, and we affirm that the data does not infringe upon the rights of any third parties.

Furthermore, we confirm that the dataset is being released under the following license: CC BY-NC-SA 4.0. his license allows others to use, share, and adapt the dataset, provided that appropriate credit is given, a link to the license is provided, and any changes are indicated.

By submitting this dataset to the NeurIPS 2024 dataset track, we agree to adhere to the terms and conditions set forth by the NeurIPS conference organizers and acknowledge that we are solely responsible for any issues related to the dataset's legal and ethical use.

# E Hosting, licensing, and maintenance plan.

As described in the Maintenance section of Appendix B, the dataset is open source and available to researchers via the Kaggle[4] platform, under the CC BY-NC-SA 4.0 license. Additionally, on kaggle we offer notebooks to reproduce the model benchmark performed in this work, and links to our zenodo repository which stores our pretrained models. All notebooks are under the Apache 2.0 license and model weights are available under the Creative Commons Attribution 4.0 International license. All the above, will be maintained actively by the authors of this work. If errors are found by users in the code or data, this can be communicated via the discussion tab available on the dataset webpage, and a newer version will be uploaded.

---

[4]`https://www.kaggle.com`

