# OpenReview forum: "MultiOrg: A Multi-rater Organoid-detection Dataset"
_NeurIPS.cc/2024/Datasets_and_Benchmarks_Track — NeurIPS 2024 Track Datasets and Benchmarks Poster_

### Official Review · Reviewer_rcwd · 2024-06-24
**A multi-rater benchmarking dataset for organoid detection**

**Rating:** 7
**Confidence:** 4
**Clarity:** Yes

**Review:**

Quantification for annotation uncertainty is important in the organoid field given the nosiness of the data. The labeling experiment was well-designed with two evaluators, and commonly used detection methods were used to benchmark the annotation result. The paper is generally well written. The only caveat I found was that the disagreement between two evaluators or discrepancy over time was not accounted for in weighing the ground truth for benchmarking evaluation. It is obvious that the purpose of annotation uncertainty evaluation should be to score the certain instances with higher weight than uncertain objects.

**Strengths:**

The amount of curated data that goes into the datasets is impressive, and the multi-label annotation experiment is well designed.

**Additional Feedback:**

None

**Correctness:**

No obvious mistakes were found in the text. In section 4.3 the authors quoted YOLOv3 as “better trade-off between performance and inference time”, but the inference time was not benchmarked in the manuscript.

**Documentation:**

The documentation and distribution of curated data and code were mentioned at the paper

**Ethics:**

I have no ethical concerns with regard to the paper.

**Limitations:**

As mentioned above, it will be nice if the annotation uncertainty itself can be used to flexibly weight the ground truth for evaluation

**Opportunities For Improvement:**

For benchmarking different DL methods, the three label sets were independently evaluated and the performance were solely taken as the mean of the three label sets. However, the significance of muliti-label experiment is to quantify annotation uncertainty of different instances, and the difference of level of confidence is not accounted for in the benchmarking method. It would be nice if the authors can come up with a new benchmarking metric where the consistently labeled organoids were more stringently evaluated towards identification error, while the inconsistently labeled incidents account less for the error as even human annotators do not agree on such results.

**Relation To Prior Work:**

Previous datasets were sufficiently cited and compared to the current dataset, which demonstrated the comprehensiveness of the current dataset construction.

**Summary And Contributions:**

In this paper, the authors constructed a dataset with multi-rater annotation for organoid detection. Tens of thousands of lung organoids were imaged and curated by two annotators at two different time points to quantify inter- and intra-person annotation uncertainly. Four methods were used to benchmark the performance of current organoid detection algorithms and revealed that the best performing method were resilient to labeling noise. The multi-label in the dataset added the dimensionality of annotation uncertainty to existing organoid data, and opened the door to method evaluation with regard to ambiguous annotations.

---

> ### Author Rebuttal · Authors · 2024-08-15
>
> We thank the reviewer for their constructive and appreciated feedback, and for acknowledging that “Quantification for annotation uncertainty is important” and “The labeling experiment was well-designed”.
>
> >For benchmarking different DL methods, the three label sets were independently evaluated and the performance were solely taken as the mean of the three label sets. However, the significance of muliti-label experiment is to quantify annotation uncertainty of different instances, and the difference of level of confidence is not accounted for in the benchmarking method. It would be nice if the authors can come up with a new benchmarking metric where the consistently labeled organoids were more stringently evaluated towards identification error, while the inconsistently labeled incidents account less for the error as even human annotators do not agree on such results.
>
> We agree that developing a consensus-based metric would be beneficial for more accurate benchmarking. We explored several options but found that addressing edge cases, such as mismatched instance counts in overlapping zones, poses a challenge (refer to Fig 2; Image ID 24; iii for an example). Therefore, we decided to use a straightforward approach by focusing on the smallest common denominator for evaluation metrics. This allows us to maintain simplicity and ensure agreement while acknowledging the limitations in handling discrepancies in human annotations. Nevertheless, please note that with the mean metric, the consensus labels get a weight of 2, compared to the inconsistent ones.
>
> >In section 4.3 the authors quoted YOLOv3 as “better trade-off between performance and inference time”, but the inference time was not benchmarked in the manuscript.
>
> Thank you for pointing out. Supplementary Table A.10 illustrates the performance and inference times of YOLOv3. We will reference it more clearly in the manuscript.

---

### Official Review · Reviewer_MGmJ · 2024-07-23
**A mutli-rater dataset of 2D microscopy images for Organoid segmentation**

**Rating:** 8
**Confidence:** 3
**Correctness:** They are correct.
**Clarity:** Yes, it is well written.

**Review:**

This paper gave a clear introduction to the new dataset it created for organoid detection. At least as meaningful is the paper's emphasis on the uncertainty quantification of multiple raters' annotation. Many people neglect the latter when creating new datasets. The design of splitting the annotations from different raters to assess the consistency over time can be good practice for future works.

**Strengths:**

Large-scale dataset of organoids with multiple labeling.

Assess the quality of multi-raters' annotations regarding variance and consistency across time.

Made the dataset publicly available by posting as a challenge on popular platforms.

**Additional Feedback:**

Overall, it is a very complete and exciting work.

**Documentation:**

Yes, there are sufficient details.

**Ethics:**

No, I didn't find anything raising the alarm.

**Limitations:**

The limitations are generally well addressed. However, even though three label sets were created, only two annotators were involved.

**Opportunities For Improvement:**

(1) Explain how quantifying the uncertainty in the annotation will help benchmark models more accurately, such as whether any metrics can be designed considering the annotation inaccuracy.

(2) Since there are already several methods for comparing multiple annotations, as mentioned in the introduction, clarifying the difference between the proposed design and previous ones would be meaningful.

(3) Show the difference visually compared to the previously published dataset.

**Relation To Prior Work:**

It clearly discussed the differences in the dataset. However, the paper needs more clarification on the difference in evaluating the uncertainty.

**Summary And Contributions:**

The paper presents the first large-scale organoid imaging dataset with multiple labels. It also quantified the label uncertainty using both inter and intra-tater uncertainties. The best model is made publicly available on the Napari platform.

---

> ### Author Rebuttal · Authors · 2024-08-15
>
> We appreciate the reviewer's constructive feedback and their support for our manuscript.
>
> >Explain how quantifying the uncertainty in the annotation will help benchmark models more accurately, such as whether any metrics can be designed considering the annotation inaccuracy.
>
> Understanding and quantifying the uncertainty in annotation plays an important role in improving the benchmarking of models in a few ways.
> For example, as shown in our previous works, the information about rater variability allows to design metrics superior to the conventional ones [https://www.melba-journal.org/papers/2023:002.html ], and reconsider the definition of the ground truth annotation [https://ieeexplore.ieee.org/document/10230497 ].
> Thank you for pointing this out, we will update the manuscript accordingly.
>
> >Since there are already several methods for comparing multiple annotations, as mentioned in the introduction, clarifying the difference between the proposed design and previous ones would be meaningful.
>
> Our proposed annotation design is indeed the simplest way to get both inter and intra-rater labels. The difference with previous works lies in that, typically, no reannotation of data by the same annotator is done. Note that providing only test^0 would be a typical design with two annotators and one label set: in that case, the train and test sets would typically be stratified by annotators.We will update the manuscript accordingly as well.
>
> >Show the difference visually compared to the previously published dataset.
>
> Thank you for this suggestion. We should add the comparison figure to the camera-ready version.

---

> > ### Comment · Reviewer_MGmJ · 2024-08-21
> > **Good response. I will remain my score:8**
> >
> > Good rebuttal.

---

### Official Review · Reviewer_xpFX · 2024-07-23
**lack of enough experts - no clear methodology on test set 3 sets labeling**

**Rating:** 3
**Confidence:** 5

**Review:**

Despite of the claim of releasing a large multi-rater dataset, the coverage of the dataset is quite limited.

- (minor limitation) Limited Organoid Diversity: The dataset focuses solely on lung organoids, limiting its generalizability to other organoid types. Expanding the dataset to include a wider range of organoids would increase its value for developing more robust and widely applicable models.

- (minor limitation) Single Modality: The use of bright-field microscopy, while common, restricts the dataset's use for exploring the potential of other imaging modalities, such as fluorescence microscopy, which might provide richer information for organoid segmentation.

- (mid level limitation) Absence of Multi-Class Labels:  While the dataset features two types of lung organoids, these are not explicitly annotated as separate classes. Incorporating this distinction would allow for investigating the performance of multi-class object detection models and could offer additional biological insights.

- (strong limitation) Limited Annotator Pool: While valuable, the use of only two expert annotators significantly limits the ability to generalize findings about inter-rater variability to a broader context.  A larger, more diverse pool of annotators would provide a more robust representation of the range of interpretations and potential disagreements in organoid labeling.

- (strong limitation) There is no a clear motivation or experimental design between the three label sets proposed. The authors indicate the potential presence of uncorrected pseudo-labels used as a starting point for annotation. That reduces the credibility of the expert's annotation. Alternatively, authors could potentially  use of sentinel test and noise annotation filtering. In practice, as there are only 2 experts, and no external comparison is it not clear the feasibility of using such annotations.

**Strengths:**

Open Availability and Benchmark: Making the dataset and code publicly available on platforms like Kaggle and Zenodo promotes transparency and encourages wider adoption and research on label uncertainty in organoid analysis. The provision of a benchmark with baseline model performance further fosters this.
Napari Integration:

Releasing a Napari plugin based on the best performing model enhances the dataset's practical utility, enabling researchers to apply it directly to their own data for organoid quantification.

 Figures effectively illustrate key points, such as the multi-rater annotation process (Figure 1) and the differences in labeling across annotators (Figure 2).

**Additional Feedback:**

as previously described.

**Clarity:**

The MultiOrg paper is generally well-written and effectively communicates its key contributions.

**Correctness:**

It is not clear that the quality of the annotations captured are reasonable. Please clarify

**Documentation:**

it is not a benchmark

**Limitations:**

The authors briefly describe the limits on incorporating multi-class labeling, but do not discuss the limitations associated with low number of experts and noise labeling.

**Opportunities For Improvement:**

- inclusion of more experts
- filtering noise annotations, and the use of sentinels for tracking annotators quality
- extending table 1 to describe number of experts adopted

**Relation To Prior Work:**

The authors describe other single and multiple modality datasets containing organoid segmentation. Table 1 summarize the discussion.

**Summary And Contributions:**

This paper introduces MultiOrg, a new multi-rater dataset for organoid detection in 2D microscopy images. The dataset contains approximately 400 images of lung organoids, with over 60,000 annotations provided by two expert annotators.

The key feature of this dataset is the inclusion of three label sets for the test data. They represent different combinations of annotations from experts one and two with different levels of "re-annotation".
These sets represent annotations from: Both annotators at an initial time point (test0).
Each annotator individually re-annotating the same images at a later time point, blinded to their original annotations (test1A and test1B).
This allows researchers to study the impact of label noise and inter/intra-rater variability on model performance.

The authors benchmark MultiOrg using four popular object detection models and demonstrate that:
Models can be robust to the level of label noise present in the initial annotations.
Different models exhibit varying precision-recall tradeoffs.

Finally, the authors release a Napari plugin called "napari-organoid-counter"  that leverages their best performing model (YOLOv3) to enable high-throughput analysis of lung organoids.

Claimed contributions:
- Release of MultiOrg: A large, publicly available multi-rater organoid dataset with over 400 microscopy images and 60,000 lung organoid annotations.
- Quantification of Label Uncertainty: Provision of a Kaggle benchmark challenge that evaluates submissions on different test label sets, allowing for the analysis of annotation biases and their impact on model training.
Benchmarking of Standard Object Detection Methods: Evaluation of four standard object detection models on MultiOrg, demonstrating performance variations based on selected annotations and highlighting the robustness of deep learning to label noise.
- Release of Napari Plugin: Development and release of napari-organoid-counter, a user-friendly Napari plugin incorporating the best-performing model for lung organoid quantification, allowing for visualization, correction, and feature extraction for downstream analysis.

---

> ### Author Rebuttal · Authors · 2024-08-15
>
> We thank the reviewer for their valuable feedback. We have clarified the motivation for this work in the general rebuttal.
>
> >Limited Organoid Diversity
>
> Please refer to our answer to reviewer uS6Q (2nd point).
>
> >Single Modality
>
> Analogously to established medical and microscopy datasets (e.g., BraTS, Table 1), we provide data from one imaging modality, which, in our case, is bright-field microscopy.
>
> Bright-field microscopy is a non-destructive method that allows for longitudinal imaging of the organoids. In contrast, fluorescence imaging would require the fixation of cells (which kills them and changes the morphology of the organoids to a certain extent) and the application of dyes that potentially interfere with cell function. In practice, this is done once to confirm phenotypes or characterize organoids and their responses to different treatments, whereas bright-field microscopy is used in high-throughput settings, where robust automatic detection methods can be very valuable.
>
> Additionally, as stated above, the main aim of our benchmark is to provide a platform for the development of label-noise-aware object detection methods. While a multi-modal dataset would certainly be a valuable resource from a biological perspective, it would introduce a higher barrier to extending the available label sets, as a multi-modal approach would introduce another level of complexity in the labeling process. Here, label noise would become a function of the image modality.
>
> >Absence of Multi-Class Labels
>
> While the multi-class problem would be interesting from a machine learning point-of-view, the underlying biological problem, namely the quantification of distal lung epithelial regeneration, doesn't require classification into the type of organoids. The main objective is to compare the number and size of organoids, as a measure of regeneration. We have therefore prioritized simplicity in our benchmark design, and also want to keep annotation noise sources disentangled.
>
> Furthermore, as highlighted above (point “Limited Organoid Diversity”), the different types of organoids ensure greater generalizability of the model, and preliminary tests have shown that our best model generalizes to other lung cell lines and a different organ (colon). The multi-class labels would not generalize, and we would like to encourage the inclusion of further organoid lines and systems under the same framework in our benchmark.
>
> >Limited Annotator Pool and inclusion of more experts
>
> Unlike most other benchmarks in the field, we provide multiple label sets. As outlined above, the required annotation experts are scarce, and the annotation is subjective.
> Nevertheless, we agree that our dataset would benefit from more annotators. Therefore, we publish it as open-access, which will hopefully motivate other experts and research groups to contribute (as it historically happened with established benchmarks, such as BraTS).
> But even in the current state, our dataset with multiple label sets from two expert annotators and inter- and intra-rater analysis, still presents a very valuable resource to the community, which is also acknowledged by the other reviewers.
>
> >experimental design and credibility of the annotations
>
> Our experimental design and the three label sets have been appreciated by reviewers MGmJ and rcwd, who wrote “The design of splitting the annotations from different raters to assess the consistency over time can be good practice for future works” and “the multi-label annotation experiment is well designed”. It is indeed the simplest way to get both inter and intra-rater labels. Note that providing only test^0 would be a typical design with two annotators and one label set: in that case the train and test sets would typically be stratified by annotators.
>
> Our annotation process follows best practices, including semi-automatic label generation, which were established for both BraTS (see above), microscopy segmentation [https://www.cell.com/cell/fulltext/S0092-8674%2822%2901465-9 ], and organoid detection [Bremer et al., 2022]. The same annotation process was used to create the dataset for another published biological study [Kastlmeier et al. 2023].
>
> Differentiating organoids from matrigel artifacts, dust, and debris in bright-field imaging is intrinsically a difficult task, which makes it suitable for studying annotation uncertainty. When comparing label sets in Table A.9, we observe that more pseudo labels were removed at time point t^1 than at t^0, but also that test^1_B was curated the least. These effects are relatively small (a few F1-score and Recall points). They highlight the subjectivity of the annotation and do not impact the credibility of those label sets. Our experts have several years of experience, and all the data included in the dataset was used for downstream biological analysis (paper in preparation). Furthermore, though all our models were trained exclusively on annotations from time point t^0, they are actually learning meaningful information as we found that they perform well, if not even better when evaluated on test^1_A and test^1_B.
>
> >filtering noise annotations, and the use of sentinels for tracking annotators quality
>
> The proposed sentinel datapoint strategy would require actual ground truth to verify that the pool of non-expert annotators is annotating properly. As we solely operate with experts, this strategy is unsuitable for our use case. The same is relevant regarding other strategies for annotation noise filtering.
>
> >extending table 1 to describe number of experts adopted
>
> Indeed, some of the previously published datasets have been annotated by more than two experts, but only one label set is provided for each image without an annotator identifier. Therefore, machine learners cannot leverage the inclusion of multiple experts. Furthermore, none of those papers carried out inter- or intra-rater studies. This is why we would refrain from including this information in Table 1.

---

> ### Comment · Reviewer_xpFX · 2024-08-26
> **clear rebuttal - still small number of experts**
>
> The authors have addressed most of my concerns. They clarified the main benefits of the proposed dataset in relation to the  available datasets. That said, I am still concerned about the small number of annotators (two) and how that reflects the proposed benchmark for quantifying uncertainty. If the task at hand is expected to vary among experts, the number of experts itself should be carefully chosen in order to reflect the expected variance.
> My updated rating is Rating: 5

---

### Official Review · Reviewer_uS6Q · 2024-07-24
**MultiOrg**

**Rating:** 6
**Confidence:** 4
**Correctness:** The claims seems to be correct.

**Review:**

MultiOrg seems to be a good dataset addition for organic detection in microscopy images, but the dataset contains a few flaws, as described in the weakness section.

**Strengths:**

- Organoid detection seems to be an important task in biomedical research, with many existing datasets available.
- The authors provided good analysis of inter-rater performance.

**Additional Feedback:**

N/A

**Clarity:**

The paper seems to be well written overall.

The use of multi-label is misleading in table 1, multi-rater or multi-annotation might be more appropriate.

**Documentation:**

Documentation seems to be sufficient.

**Ethics:**

No ethical concerns.

**Limitations:**

The limitations are addressed in the paper.

**Opportunities For Improvement:**

- The key contribution of this dataset is having multiple annotators to label the same data, but this is only in the testing set. This may be good for evaluation, but as the authors pointed out, this does not allow the community to leverage multiple annotations to train detection models.
- MultiOrg is limited to a single organ system.
- Annotation includes only detection labels, without instance masks for organoids.
- Benchmarks are limited to convolutional based methods. Inclusion of more recent, publicly available methods such as [DETR](https://github.com/facebookresearch/detr), [MedSAM](https://github.com/bowang-lab/MedSAM), or [Cellpose](https://www.cellpose.org) may boost the impact of the paper.

**Relation To Prior Work:**

Prior works are addressed.

**Summary And Contributions:**

The authors proposed MultiOrg, an organoid dataset designed for object detection in high-throughput image analysis. The dataset includes over 400 microscopy images with curated annotations of more than 60,000 organoids by three label sets by two annotators. The authors also included benchmarks of existing object detection methods on the dataset.

---

> ### Author Rebuttal · Authors · 2024-08-15
>
> We thank the reviewer for their valuable feedback and for acknowledging that “Organoid detection seems to be an important task in biomedical research” and that “The authors provided good analysis of inter-rater performance”. We have clarified the motivation for this work in the general rebuttal. Here is our response to the specific points raised above:
>
> >The key contribution of this dataset is having multiple annotators to label the same data, but this is only in the testing set. This may be good for evaluation, but as the authors pointed out, this does not allow the community to leverage multiple annotations to train detection models.
>
> While uncertainty in labeling is omnipresent in various datasets, having several label sets in the training set is often very expensive and, therefore, quite unrealistic: a typical practical training setup is based on one set of labels. Thus, we want to incentivize the community to benchmark machine learning approaches that would be trained on a single set of labels (the most standard setup), but with the possibility of evaluating the robustness of trained models against the label uncertainty. We also indicate the origin of the annotations in the training set in a pseudonymized way, enabling users to develop and test methods accounting for annotator-specific label noise.
>
> >MultiOrg is limited to a single organ system.
>
> It is true that MultiOrg consists of organoids derived from a single organ. However, it is important to note that lung organoids are becoming increasingly important due to the rise in respiratory infections, including the COVID-19 pandemic.
>
> Furthermore, we have carefully included diversity in our dataset through several study setups and cell lines to ensure good generalization. We have done preliminary tests of our model on different lung cell types not present in the dataset, from both human and mouse organoids, and seen that it generalizes quite well. We also tested it successfully on colon organoids (preliminary data that cannot be shared) and speculate that it can be used for all organoids with similar shape and size. We will mention this in the final version of the manuscript.
>
> Finally, as we appreciate the added value of multiple organoid types, the extension to other organoid types is on our roadmap. Here, the open-access release of our dataset on platforms (Kaggle and Zenodo), combined with the functionality to automatically label using Napari, is expected to encourage the inclusion of other organoid systems under the same framework.
>
> >Annotation includes only detection labels, without instance masks for organoids.
>
> Bounding box annotations are much more time-efficient to create than instance segmentations and suffice for most practical applications with organoids. Indeed, organoid detection is used to monitor high-throughput cell cultures, i.e., primarily for counting organoids and sometimes for size evaluation and tracking. For size evaluation, bounding boxes are usually sufficient: organoids are typically spherical objects, allowing for straightforward size extrapolation.
>
> Furthermore, we would like to argue that the detection of the objects in those images is the challenging part from a machine learning point-of-view. Once the detection has been done, the segmentation can be obtained from pre-trained segmentation models. To support this claim, we tested a subsample of our test data using SAM [https://ai.meta.com/research/publications/segment-anything/ ] and providing the bounding box annotations as prompts to the model. As can be seen in the attached pdf the segmentation results obtained are of high quality. We will convey this point in the final version of our manuscript.
>
> >Benchmarks are limited to convolutional based methods. Inclusion of more recent, publicly available methods such as DETR, MedSAM, or Cellpose may boost the impact of the paper.
>
> Thank you for pointing this out.
> We initially implemented DETR with standard hyper-parameters for our dataset, but the training was quite unstable, and the performance much worse than other models. We therefore decided not to report the scores in the manuscript.
>
> Furthermore, we provide a detection benchmark, and MedSAM and Cellpose are segmentation-based approaches. They did not work well out-of-the box and would require segmentation labels for fine-tuning to be useful.
>
> We will clarify those points in the final version of the manuscript.

---

### Author Rebuttal · Authors · 2024-08-15

We thank the reviewers and area chairs for their efforts in improving our manuscript. Based on their feedback, we would like to clarify the motivation behind our dataset, and will make some adjustments to the manuscript to communicate it better.

The biomedical image analysis community faces several unique challenges compared to conventional computer vision research:
1. Scarcity of Data. Acquiring biomedical images is costly and requires specialized equipment, expertise, and extensive pre- and post-processing.
2. Scarcity of Annotations. Image annotation is time-intensive and requires experts with years of training. Despite this expertise, annotation is highly subjective in most biomedical contexts.
3. Data Aggregation Difficulties. Large datasets are hard to aggregate due to institutional boundaries, data protection regulations, and varied acquisition procedures.

Key datasets in the medical field, such as BraTS [https://arxiv.org/abs/1811.02629 ], typically gather hundreds of patients, and microscopy datasets, such as [https://www.nature.com/articles/s41592-024-02233-6 ]  typically gather thousands of cells (see also Table 1). Those are accompanied by one label set, often created by multiple annotators from different institutions, potentially without a clearly defined common labeling procedure [https://www.nature.com/articles/s42256-023-00625-5 ].

Our benchmark aims at embracing the subjectivity of data annotation. We provide a training dataset with a single set of labels from two annotators. Offering additional labels would be ideal, but it's atypical due to high annotation costs.

We also indicate the origin of the annotations in a pseudonymized way, enabling users to develop methods accounting for annotator-specific label noise.

For the test set, we provide multiple expert annotations to reflect labeling subjectivity. Usually, developers aim for similarity with a single label set, but perfect overlap with such may lead to overfitting on annotation noise and not translate to better models [https://arxiv.org/abs/2301.00243 ].

By making this dataset and benchmark available on Kaggle, we want to encourage the development of methods that account for annotation noise while maintaining practicality, as the resources needed to generate multiple label sets do not exist for most biomedical problems. Moreover, by providing the option for running model inference through the well-established software napari [https://napari.org/0.4.15/index.html ] we enable access to DL models for non-ML experts.

---

### Decision · Program_Chairs · 2024-09-26

**Decision:**

Accept (Poster)

**Comment:**

**Submission #1646**: "MultiOrg: A Multi-rater Organoid-detection Dataset"

**Recommendation: Accept (Poster)**

**AC Comment/Note**: Lung cell organoids are definitely somewhat outside my area of expertise; however, manually generated spatial (intra-) image annotations are something I am quite familiar with. The overall work seems solid, and despite the (very) small number of annotators (which for a project of this size is still remarkable), I believe this will be a useful resource for the community, given the overall positive feedback.

**Score note**: One of the reviewers (initial score 3) replied to the rebuttal with the intention to raise the score to 5 (bringing the average to 6.5), which was however not executed on the platform. Given the overall (presumed final) average of 6.5, and the reviewer being quite happy with the rebuttal, I believe this indicates general approval with the submission.

**Summary**: The authors present a dataset of 411 bright-field microscopy images showing wells of lung cell organoid together with over 60,000 annotated organoids detected within these images. The annotations are generated by two expert annotators, and provided across different sessions, allowing for at least some more complex analysis w.r.t. annotation reliability, giving future benchmark efforts a chance for some initial variability estimates (w.r.t. annotator influence). Together with the dataset, the authors also release a Kaggle challenge (allowing for further community provided annotations), as well as a detection model packaged into a Napari plugin, allowing for applying the model trained by the authors to novel data.

The reviewers highlight the application (biomedical research, in which organoids have become an important aspect) and the scale as well as the aspect of multiple annotators (even if only two) providing the annotations, allowing for more complex estimations.